

# Vehicular volatile organic compounds (VOCs)-NO$_x$-CO emissions in a tunnel study in northern China: emission factors, profiles, and source apportionment

Congbo Song[1], Yan Liu[1], Shida Sun[1], Luna Sun[1], Yanjie Zhang[1], Chao Ma[1], Jianfei Peng[2], Qian Li[1], Jinsheng Zhang[1], Qili Dai[1], Baoshuang Liu[1], Peng Wang[2], Yi Zhang[1], Ting Wang[1], Lin Wu[1], Min Hu[3], and Hongjun Mao[1]

[1]Center for Urban Transport Emission Research & State Environmental Protection Key Laboratory of Urban Ambient Air Particulate Matter Pollution Prevention and Control, College of Environmental Science and Engineering, Nankai University, Tianjin, 300071, China
[2]Zachry Department of Civil Engineering, Texas A and M University, College Station, TX, 77845, USA
[3]Laboratory of Environmental Simulation and Pollution Control, College of Environmental Sciences and Engineering, Peking University, Beijing, 100871, China

**Correspondence:** Hongjun Mao (hongjunm@nankai.edu.cn)

**Abstract.** Vehicular emission is a key contributor to ambient volatile organic compounds (VOCs) and NO$_x$ in Chinese megacities. However, the information of real-world emission factors (EFs) for a typical urban fleet is still limited, hindering the development of a more reliable emission inventory in China. Based on a more-than-two-week (August 8-24, 2017) tunnel test in urban Tianjin in northern China, and on the use of a statistical regression model, the Positive Matrix Factorization (PMF) receptor model, and the Calculate Emissions from Road Transport (COPERT) IV model, characteristics of vehicular VOCs-NO$_x$-CO emissions were analyzed systematically. The fleet-average EFs (pollutant: downslope, upslope, and overall in mg km$^{-1}$ veh$^{-1}$) were estimated respectively as follows: (NO: 61.92±72.46, 158.58±73.48, 97.52±69.84), (NO$_2$: 16.52±11.49, 23.98±20.14, 15.86±9.38), (NO$_x$: 79.45±78.43, 181.22±88.29, 116.56±77.61), and (CO: 269.96±342.38, 577.76±382.22, 344.67±250.01). The EFs of NO-NO$_2$-NO$_x$ and CO from heavy-duty vehicles (or diesel vehicles) were differentiated from light-duty vehicles (or gasoline vehicles). The ratios (v/v) of NO$_2$ to NO$_x$ in the primary vehicular exhaust were approximately 0.18±0.09, 0.10±0.22 and 0.10±0.05 for downslope, upslope, and the entire tunnel, respectively. The fleet-average EF of the 99-target non-methane VOCs (NMVOCs) was 40.56±12.18 mg km$^{-1}$ veh$^{-1}$, lower than the previous studies in China. The BTEX (benzene, toluene, ethylbenzene, $p$-xylene, $m$-xylene and $o$-xylene) levels decreased by approximately 79% when emission standards increased from China I to China V. The source profiles of NMVOCs from the tailpipe and evaporative emissions were resolved by the PMF model. The evaporative emissions accounted for nearly one-half of the total vehicular VOC emissions, indicating that evaporative and tailpipe emissions contributed equally to VOC emissions. The relative contributions of evaporative NMVOC emissions to total vehicular NMVOC emissions are temperature-dependent with the average increasing ratio of 7.55% °C$^{-1}$. The primary emission ratio (ER, m/m) of VOCs/NO$_x$ was approximately 2.04, suggesting that vehicular NO$_x$ and VOCs can be co-emitted with a proper ER. According to the vehicular ERs of VOCs/NO$_x$ in Tianjin (2000-2016) and China (2010-2030), as even more stringent emission standards are implemented in the future, the O$_3$ chemical regimes were



likely to be VOCs-limited (i.e., 8:1 threshold) for cities or regions where VOCs and $NO_x$ emissions are dominated by vehicular exhaust. Our study enriched the database on the fleet-average emission factors of on-road vehicles for emission inventory, air quality modeling, and health effects studies, provided implications for following $O_3$ control in China from the view of primary emission, and highlighted the importance of further control of evaporative emissions.

## 5  1  Introduction

Volatile organic compounds (VOCs) and $NO_x$, largely emitted from fossil-fuel-powered vehicles in urban areas, are always associated with severe haze, tropospheric ozone ($O_3$) episodes and human health risks (Lelieveld and Pöschl, 2017; Li et al., 2017; Song et al., 2017b; Peng et al., 2017; Kelly and Zhu, 2016; Liu et al., 2015b; Guo et al., 2014; Parrish and Zhu, 2009). VOCs are precursors of secondary organic aerosol (SOA) and tropospheric $O_3$ (Huang et al., 2015; Atkinson and Arey, 2003;

Carter, 1994). $NO_x$ controls have been reported to be more beneficial than $SO_2$ controls for improvement of $PM_{2.5}$ air quality in northern China (Song et al., 2017b; Cheng et al., 2016). Vehicular activity is the most significant contributor to ambient VOCs in Chinese megacities, such as Beijing (Wang et al., 2016), Tianjin (Liu et al., 2016; Han et al., 2015), Shanghai (Cai et al., 2010), and Guangzhou (Wu et al., 2016). Vehicle emissions have also been recognized as a major source of $NO_x$ in northern China (Jing et al., 2016; He et al., 2016; Tan et al., 2016). Establishing reliable emission inventories for VOCs, $NO_x$

and CO are crucial for the photochemical study and simulating the complex atmospheric environment changes (Zhang et al., 2018a; Wang et al., 2017). Although emission factors (EFs) of VOCs, $NO_x$ and CO from individual vehicle have been widely tested by the portable emission measurement system (PEMS) (Zhang et al., 2016b), information on real-world EFs for a typical urban fleet is still limited (Song et al., 2018), hindering the development of a more reliable emission inventory in China.

It is known that there are two types of VOC emissions from vehicles: tailpipe emissions and evaporative emissions. Tailpipe

emissions have been well-controlled across the global, but vehicular evaporative emissions have been largely neglected in China. With the control of tailpipe exhaust emissions, the contribution of evaporative emissions to vehicular VOC emissions is rapidly increasing. Vehicular evaporative emissions can be generally grouped into hot soak, diurnal, permeation, refueling process, and running loss (Yue et al., 2017; Liu et al., 2015a). Source profiles and EFs of VOC from hot soak, diurnal, permeation and refueling process have been well characterized in laboratories, but VOC emissions from running loss are less understood

because no facility in China could accommodate the test procedures or track-based tank temperature profile generation (Liu et al., 2015a). Tunnel test might be the most idealized measure to estimate primary vehicular VOC emissions including both tailpipe and evaporative emissions (mostly running loss) without oxidation degradation by ultraviolet (UV) light (Kowal et al., 2017). Source apportionment methods could be used to apportion tailpipe and evaporative VOC emissions (Gertler et al., 1996). Currently, the relative contributions of tailpipe and evaporative emissions to vehicular VOC emissions in China remain poorly

quantified, though evaporative emissions are a major source of VOC emissions in China (Liu et al., 2015a).

In addition, few studies have been performed to determine the emission ratio (ER) of VOCs to $NO_x$ (Ehlers et al., 2016), which plays a critical role in identifying the $O_3$ chemical regimes for reducing atmospheric secondary pollution (Edwards et al., 2014; Liu et al., 2013; Schnell et al., 2009). Although all emission control strategies have benefits in the reduction



of $O_3$ precursors, inappropriate reduction ratios of VOCs to $NO_x$ can lead to an increase in surface $O_3$ (Liu et al., 2013). The well accepted theory is that a transition from a VOCs-limited to $NO_x$-limited regime appeared at a VOCs/$NO_x$ ratio of approximately, e.g., 8:1, for VOCs expressed as the concentration of carbon atoms (Seinfeld, 1989). Thus far, the vehicular emission ratio of VOCs/$NO_x$ remain under study, though vehicular emission is the key contributor to both ambient VOCs and

$NO_x$ in Chinese megacities. The $NO_x$ emitted from on-road vehicles is released into the atmosphere as a mixture of NO and $NO_2$ (Song et al., 2018). $NO_2$, instead of $NO_x$, is among the criteria pollutants according to the World Health Organization (WHO) Air Quality Guidelines because of associated adverse health effects. Understanding the ratio of $NO_2$ to $NO_x$ in primary vehicular exhaust ($f_{NO_2}$) is necessary for photochemical production of $O_3$ and vehicle emission control strategies (Ehlers et al., 2016; Jenkin, 2014; Clapp and Jenkin, 2001).

Beginning in the 1990s, China has implemented vehicular emission control programs, including controls for new vehicles and in-use vehicles, improvements in fuel quality (e.g., the promotion of unleaded gasoline and low-sulfur fuels), and promotion of alternative fuels and incentives for electric vehicles (Wu et al., 2017). Emission standards have been upgraded from China I (equals to Euro I) in 2001, to China V in 2015. China is experiencing a rapid improvement in vehicle emission control technologies that have impacted vehicular emissions. However, information on real-world EFs for a typical urban fleet remains

limited, particularly for VOCs including tailpipe and evaporative emissions. The fleet-average EFs measured from a inner urban tunnel are EFs of typical driving cycles in urban areas. The overall objectives of this study were to: (1) update real-world local EFs of VOCs, NO-$NO_2$-$NO_x$, and CO from detailed vehicular types in Tianjin in northern China; (2) investigate the primary $NO_2$/$NO_x$ ratio and VOCs/NOx ratio for on-road vehicular emissions; (3) explore the source profiles of tailpipe and evaporative emissions and their relative contributions to vehicular VOC emissions; and (4) diagnose the trends of the vehicular

EFs of VOCs-$NO_x$-CO in China. The results from this study enriched the database on the fleet-average emission factors of on-road vehicles for source apportionment, emission inventory, air quality modeling, and health effects studies, and provided new insights into tailpipe and evaporative VOC emissions.

## 2   Materials and methods

### 2.1   Tunnel description

The measurement campaign was conducted from August 8th to August 24th, 2017, in the Wujinglu (WJL) tunnel (117°12′15″, 39°8′31″), an urban tunnel in the city center of Tianjin in northern China. In the WJL tunnel (as shown in Fig.1), 14866±900 (mean±standard deviation) vehicles passed through per day during the measurement campaign. As there is a 40 km h$^{-1}$ speed limit in the tunnel, the average traffic speeds during the study period were 35.6±1.7 km h$^{-1}$ at the entrance of the tunnel, and 38.7±2.1 km h$^{-1}$ at the tunnel exit. The fleet in the WJL tunnel was mainly comprised of light-duty passenger vehicles

(LDPVs), which accounted for 96.33±0.69% of the total vehicles. Gasoline vehicles (GVs) was the majority (94.24±0.32%) in the fleet. Measurements were continuously conducted at inlet (1#, 45 m from the tunnel entrance), middle (2#, 605 m from the tunnel entrance), and outlet (3#, 975 m from the tunnel entrance) sites on the same side in the north bore. The validated





lengths for the downslope (1#-2#, gradient of approximately -4%), upslope (2#-3#, gradient of approximately 4%), and overall (1#-3#) of the tunnel are 560, 370, and 930 m, respectively. The cross-sectional area of the tunnel is approximately 54 m$^2$. There was no fresh air supply throughout the bores; therefore, dilution of air pollutants was eliminated. Longitudinal jetting ventilation fans along the ceiling throughout the tunnel were inactive during the sampling periods. Ventilation was thus only

induced by the flow of traffic through the tunnel and prevailing winds. As the traffic light control is at least 250 m from the entrance of the tunnel, cold start vehicular emissions were negligible.

## 2.2 Measurements and analytic methods

Meteorological data including temperature (T), relative humidity (RH), wind speed (WS), wind direction (WD), and atmospheric pressure (P) were measured using VAISALA WXT520 (Helsinki, Finland) automatic weather stations with a time

resolution of 1 min. The actual volumetric flow rates induced by the vehicular fleet and the prevailing winds in the tunnel were continuously measured using ultrasonic gas flowmeters (Flowsick-200 SICK MAIHAK, Germany) with a time resolution of 1 min, which have also been used in previous tunnel tests (Song et al., 2018; Zhang et al., 2018b; Imhof et al., 2006).

Traffic count and vehicular speed were continuously monitored using roadside laser loop detectors (AxleLight RLU11) that were installed at both the inlet (1#) and middle (2#) of the WJL tunnel. The vehicles could be grouped into 14 categories based

on wheelbases and axle-number. A high-definition vehicular license plate recognition system was installed at a pedestrian overpass approximately 115 m from the exit of the tunnel. Additionally, video footage was also recorded for data validation and review. The license plates of vehicles passing through the tunnel during the measurement campaign were matched with the registered vehicle database (until August 2017) of Tianjin to obtain vehicular types, in-use fuel types, emission standards, and other information regarding the vehicles. Vehicular classification results were obtained from the license plates in this study,

instead of those from the loop detectors. The fleet was classified into gasoline vehicles (GVs), diesel vehicles (DVs), and alternative-fuel vehicles based on the in-use fuel types. On-road vehicles were divided into light-duty vehicles (LDV), and heavy-duty vehicles (HDV). The fleet comprised China I to China V vehicular emission standards.

The NO-NO$_2$-NO$_x$, O$_3$, and CO were measured by micro-monitoring station (Environnement S.A, France), which integrated AC32M module for NO-NO$_2$-NO$_x$ analyzer, O342 module for O$_3$ analyzer, and CO12 module for CO analyzer (Song et al.,

2018). The AC32M measurement module was used to continuously measure NO-NO$_2$-NO$_x$, which operates on the principle that NO will emit light (chemiluminescence) in the presence of highly oxidizing O$_3$ molecules. The O342 measurement module (specifically recording low concentrations) uses the principle of O$_3$ detection by absorption in ultraviolet light. The continuous CO12 measurement module uses the principle of detection by absorption in infrared light. The minimum detectable limits for the NO-NO$_2$-NO$_x$, O$_3$, and CO analyzers are 0.4ppb, 0.4ppb, and 0.05 ppm, respectively. We regularly maintained, calibrated,

and cleaned the instruments to ensure that the measurements were reliable. Datasets for NO-NO$_2$-NO$_x$, O$_3$ and CO were available as 1 min values, which were then aggregated to 1 h mean values. Additionally, ambient concentrations of air pollutants (PM$_{2.5}$, NO-NO$_2$-NO$_x$, O$_3$ and CO) during the measurement periods were collected from national air quality monitoring sites (NAQMS) (Song et al., 2017b).





The non-methane VOCs (NMVOCs) samples were collected at 3-h intervals both in the inlet (1#) and middle (2#) of the tunnel in 3.2-L pre-evacuated stainless-steel canisters (Entech Instruments, Inc., Simi Valley, CA, USA) at a constant flow rate of 17.8 mL min$^{-1}$ on August 10-11 and August 19-20, 2017 (two weekdays and two weekend days). The NMVOCs samples were collected at 3-h intervals (00:00-03:00, 03:00-06:00, 06:00-09:00, 09:00-12:00, 12:00-15:00, 15:00-18:00, 18:00-21:00,

and 21:00-00:00 LT) on each sampling day. The samples in the canisters were drawn into the pre-concentrator from one of the two-channel samples. A total of 99 target NMVOC species (29 alkanes, 12 alkenes, 16 aromatics, 28 Halocarbons, and 14 OVOCs (oxygenated VOCs)) were analyzed using a high-resolution gas chromatography-flame ionization detector (GC-FID) and gas chromatography-mass selective detector (GC-MS). Details regarding the laboratory analysis of the NMVOCs can be found in our previous study (Zhang et al., 2018b).

## 2.3   Emission factor calculation

The average EFs for vehicles traveling through the tunnel were estimated using the following formula which has been widely used in previous studies (Song et al., 2018; Fang et al., 2018; Zhang et al., 2018b; Gertler et al., 1996; Pierson et al., 1996; Pierson and Brachaczek, 1983).

$$EF_{fleet} = \frac{(C_{Outlet} - C_{Inlet}) \times M \times A \times v \times T}{V_m \times N \times L} \tag{1}$$

where, EF (mg km$^{-1}$ veh$^{-1}$) is the fleet-average emission factor. $C_{Outlet}$ and $C_{Inlet}$ (ppm) are the air pollutants concentrations at the outlet and inlet of the tunnel, respectively. $M$ (g mol$^{-1}$) is the relative molecular mass of the air pollutant. $A$ (m$^2$) is the tunnel cross section area in m$^2$ (54 m$^2$). $v$ (m s$^{-1}$) is the air velocity parallel to the tunnel measured by the ultrasonic gas flowmeters. $V_m$ is the standard molar volume (22.4 L mol$^{-1}$ in this study). $N$ (veh) is the traffic count traveling through the tunnel during the time interval ($T$ = 3600 s). $L$ (m) is the validated length between the two sampling sites (downslope: 560 m,

upslope: 370 m, and overall: 930 m).

NO at the inlet of the tunnel can be converted to NO$_2$ in the presence of O$_3$ transported from outside of the tunnel mainly by the following reaction:

$$NO + O_3 \rightarrow NO_2 + O_2 \tag{R1}$$

Thus, there were three NO$_2$ sources measured at the outlet of the tunnel: ambient NO$_2$ carried into the tunnel due to the

piston effect, the transformation of NO, and the primary vehicular NO$_2$. The ratios of the EFs of NO$_2$ to EFs of NO$_x$ from on-road vehicles were defined as follows:

$$Ratio1 : ER_{NO_2/NO_x} = \frac{[NO_2]_{Outlet} - [NO_2]_{Inlet}}{[NO_x]_{Outlet} - [NO_x]_{Inlet}} \tag{2}$$

$$Ratio2 : ER_{NO_2/NO_x} = \frac{([NO_2]_{Outlet} - [NO_2]_{Inlet}) - ([O_3]_{Inlet} - [O_3]_{Outlet})}{[NO_x]_{Outlet} - [NO_x]_{Inlet}} \tag{3}$$





where Ratio 1 is the ratio of the primary emitted and transformed $NO_2$ to primary emitted $NO_x$ by volume. Ratio 2 is the ratio of the primary emitted $NO_2$ excluding the $O_3$ titration reaction (R1) to primary emitted $NO_x$ by volume. Thus, the $C_{Outlet} - C_{Inlet}$ in equation (1) were $([NO]_{Outlet} - [NO]_{Inlet}) + ([O_3]_{Inlet} - [O_3]_{Outlet})$ for NO, and $([NO_2]_{Outlet} - [NO_2]_{Inlet}) - ([O_3]_{Inlet} - [O_3]_{Outlet})$ for $NO_2$ when estimating the primary EFs of the NO and $NO_2$ for on-road vehicles (Song et al., 2018).

5    A regression method was used to differentiate between the EFs of HDV and LDV, and DVs and GVs. The regression model (Song et al., 2018; Colberg et al., 2005; Gertler et al., 1996; Pierson et al., 1996) is as follow:

$$EF = \alpha + \beta \times pCV + \epsilon \tag{4}$$

where $EF$ is the fleet-average emission factor, $\alpha$ is the EF of the LDV (or GVs), $\alpha + \beta$ is the EF of the HDV (or DVs), pCV is the proportion of the specific vehicle categories (HDV or DVs), and $\epsilon$ is the random error.

10  **2.4    Ozone formation potential**

NMVOCs play a significant role in the formation of troposphere $O_3$ (Edwards et al., 2014; Seinfeld, 1989). Ozone formation potential (OFP) is widely used to assess VOC species involved in $O_3$ formation. The maximum incremental reactivity (MIR) factors from Carter (1994) were used to calculate the OFP from vehicle emissions. The MIR factors are in units of grams of $O_3$ per gram of NMVOCs; therefore, the EF of potential $O_3$ for individual NMVOC could be estimated by the MIR coefficient 15  times the EF for each NMVOC species (Zhang et al., 2018b; Cui et al., 2018; Ho et al., 2009).

**2.5    PMF receptor model description**

Positive matrix factorization (PMF) is an advanced receptor model that decomposes a matrix of sample data ($X$) into two matrix, source contribution matrix ($G$) and source matrix ($F$), based on observations at the sampling site. The PMF model can be expressed as follows (Paatero and Tapper; Paatero, 1997):

20  $$X_{ij} = \sum_{k=1}^{p} g_{ik} f_{kj} + e_{ij} \tag{5}$$

where, $X_{ij}$ is the $j$th compound concentration of the $i$th sample, $g_{ik}$ is the contribution of the $k$th source to the $i$th sample, $f_{kj}$ is the source profile of $j$th compound in the $k$th source, $e_{ij}$ is the residual matrix for the $j$th compound in the $i$th sample, and $p$ is the total number of independent sources.

The elements ($g_{ik}$ and $f_{kj}$) are constrained to non-negative values. The task of PMF is to calculate the minimum value $Q$, 25  as follows:

$$Q(E) = \sum_{i=1}^{n} \sum_{k=1}^{m} (\frac{e_{ij}}{\sigma_{ij}})^2 \tag{6}$$

where, $\sigma_{ij}$ is the uncertainty in the $j$th compound for the $i$th sample. In this study, EPA PMF 5.0 was used for distinguishing tailpipe and evaporative sources in vehicular NMVOC emissions. More detailed PMF operations were reported in previous studies (Liu et al., 2018; Zheng et al., 2018; Liu et al., 2016).



## 2.6 COPERT IV model

The inter-annual trends of vehicular emission were estimated by the Calculate Emissions from Road Transport (COPERT) IV model. Because Chinese vehicular emission standards are similar to Europe's, the COPERT model, used to estimate emission inventories in European countries, could also be used to calculate vehicle emissions in China (Sun et al., 2016; Jing et al., 2016). The COPERT IV model is an "average speed" model, and the assessment of vehicle emission factors relies on speed-dependent equations. The COPERT IV model has been widely used to establish vehicular emission inventories in China (Sun et al., 2016; Jing et al., 2016).

## 3 Results and discussion

### 3.1 Measurements data overviews

Variations in concentrations of the air pollutants ($NO$-$NO_2$-$NO_x$, $O_3$, and $CO$), flow velocity (measured by the Flowmeter), and traffic flow (measured by the loop detector) during the measurement campaign are shown in Fig.A1-A2. Diurnal cycles in the concentrations of air pollutants, flow velocity, and traffic flow were found at both the inlet (1#), middle (2#) and outlet (3#) of the tunnel, implying that the air pollutants at near-traffic sites are generally affected by traffic activities. The WS measured in the inlet of the tunnel were mainly influenced by the prevailing wind in the ambient air, with the Pearson's correlation coefficient (Pearson's r) of 0.23 ($p$<0.001). The velocity flow measured in the middle (2#) of the tunnel using the flowmeters (Fig.A2a) were more representative of the actual volumetric flow rate, which showed a good correlation with traffic flow with explained variance of 79.2% (Fig.A2c). Thus, the velocity flow data measured using the flowmeters were used to estimate the fleet-average EFs using equation (1).

Average meteorological factors and the concentrations of air pollutants during the measurement campaign are summarized in Table A1. To some degree, the air pollutant concentrations measured at the inlet of the tunnel can be regarded as roadside air pollution characteristics. The average $NO$, $NO_2$, $NO_x$, $O_3$ and $CO$ concentrations at the inlet (roadside) were 6.15±7.60, 5.01±2.56, 5.49±2.87, 0.36±0.20, and 2.31±0.66 times greater than those in the ambient air, respectively. The average $NO$, $NO_2$, $NO_x$, $O_3$ and $CO$ concentrations at the middle of the tunnel were 39.81±30.81, 8.02±5.12, 12.00±7.63, 0.02±0.04, and 2.62±0.77 times greater than those in the ambient air, respectively. The average $NO$, $NO_2$, $NO_x$, $O_3$ and $CO$ concentrations at the outlet of the tunnel were 114.84±71.79, 9.01±6.32, 23.95±17.12, 0.002±0.01, and 2.98±1.04 times greater than those in the ambient air, respectively. Relatively higher air pollutant concentrations throughout the tunnel (inlet, middle and outlet) were observed as compared with those in the ambient air, except for $O_3$ because of titration reaction. The average NMVOC concentrations at the inlet and middle of the tunnel were 73.46±26.70 ppb and 104.31±23.06 ppb, approximately 1.50 and 2.13 times greater than the average concentration of the ambient air NMVOCs in Tianjin in August (Liu et al., 2016), respectively, suggesting that traffic activity is a key contributor to near-road VOCs. Adverse health effects have been reported to be associated with the traffic-related air pollution exposure (Sinharay et al., 2017; Cepeda et al., 2017; Chen et al., 2017; Gauderman et al., 2015; Hoek et al., 2002). Our results suggested that the air pollution levels at roadside and tunnel micro-environments were





far higher than those measured in the ambient air (NAQMS), indicating high traffic-related health risks because of the high population density living near major roads. The traditional assessment of the health burden associated with exposure to air pollution might be underestimated due to the low spatial resolution of exposure assignment (Song et al., 2017a), and should be carefully adjusted to account for the high traffic-related exposure risks.

The highest $NO_2/NO_x$ (v/v) were observed in the ambient air (0.85±0.08), followed by the inlet (0.79±0.11), middle (0.58±0.15) and outlet (0.34±0.11). The $NO_2/NO_x$ decreased as the measurement site moved closer to vehicular emission sources. During the study period, $NO_2$ constituted the dominant fraction of $NO_x$ at all three sites, although NO was the dominant proportion of $NO_x$ from the vehicular emissions (Wild et al., 2017; Simmons and Seakins, 2012; Soltic and Weilenmann, 2003). Our results differed from previous studies, which reported that NO constituted the dominant fraction of $NO_x$ in the
United Kingdom (UK) (Carslaw, 2005) and Korea (Pandey et al., 2008). However, the $NO_2/NO_x$ ratio was reported to be 0.60 in Beijing (Wang et al., 2015) and 0.61 in Langfang (Song et al., 2016), two neighboring cities of Tianjin. The high fractions of $NO_2$ in $NO_x$ at the three sites were likely because of the strongly oxidizing atmosphere in the Beijing-Tianjin-Hebei region. The $O_x$ ($NO_2+O_3$) levels were highest at the outlet (99.53±30.42 ppb), followed by the middle (90.03±28.42 ppb), inlet (77.17±17.21 ppb), and ambient air (60.17±22.38). The different pollution characteristics of the photochemical oxidants in
the ambient, inlet, middle and outlet of the tunnel might be attributed to the oxidant partition induced by the $NO_x$ variability (Jenkin, 2014; Clapp and Jenkin, 2001). Estimation of the primary on-road vehicular $NO_2/NO_x$ emission ratios could help to understand the oxidation of NO to $NO_2$ in the traffic micro-environment. $VOCs/NO_x$ ratios have been widely used to determine the $O_3$ chemical regimes (Seinfeld, 1989; Zou et al., 2015). Seinfeld (1989) reported that $O_3$ formation transited from VOCs-limited to $NO_x$-limited when the $VOCs/NO_x$ ratio was greater than 8:1 for the VOCs expressed as the concentration of
carbon atoms (ppbC). The inlet (3.43±1.50 ppbC/ppbv) and outlet (2.74±0.90 ppbC/ppbv) of the tunnel were VOCs-limited regions during the measurement campaign according to the $VOCs/NO_x$ ratios (i.e., the 8:1 threshold) (Seinfeld, 1989).

### 3.2   Emission factors of $NO$-$NO_2$-$NO_x$ and CO

As shown in Fig.2, we estimated fleet-average EFs under road conditions of downslope (1#-2#), upslope (2#-3#), and overall (1#-3#). The fleet-average EFs in this study (pollutant: downslope EF, upslope EF, and overall EF in mg km$^{-1}$ veh$^{-1}$)
were estimated, respectively, as follows: (NO: 61.92±72.46, 158.58±73.48, 97.52±69.84), ($NO_2$: 16.52±11.49, 23.98±20.14, 15.86±9.38), ($NO_x$: 79.45±78.43, 181.22±88.29, 116.56±77.61), and (CO: 269.96±342.38, 577.76±382.22, 344.67±250.01). The fleet-average upslope EFs of NO, $NO_2$, $NO_x$, and CO were approximately 2.6, 1.4, 2.3, 2.1 times greater than those of the downslope, suggesting the EFs of $NO$-$NO_2$-$NO_x$ and CO are greatly affected by road gradients. These results were comparable with Chang et al. (2008)'s study which noted that upslope (gradient of around 1.3%) pollutant EFs for $PM_{2.5}$, $PM_{10}$, CO, $NO_x$,
$SO_2$ are twice as large as those of the downslope. The EFs of $NO$-$NO_2$-$NO_x$ and CO in this study were compared to those from other tunnel studies in China (Table 1). The downslope (upslope) EFs (mg km$^{-1}$ veh$^{-1}$) of $NO_x$ and CO improved from 145±67 (331±166) and 910±470 (1470±630) in 2006 (Chang et al., 2008) to 79±78 (181±88) and 270±342 (578±382) in 2017. In addition, our results were the lowest compared with other tunnel studies in China mainly because of the low propor-





tion of heavy-duty diesel vehicles (0.35±0.35%) in the fleet and the strict enforcement of vehicular emission control strategies (China IV: 40.20±0.85%, China V: 47.76±1.29%).

As shown in Fig.2, the average EFs of NO, $NO_2$, $NO_x$, and CO showed similar diurnal variations, with the highest values before dawn (00:00-05:00 LT). As tested by Zhang et al. (2015) in Guangzhou in 2014, the EF of $PM_{2.5}$ also varied diurnally,

with the highest values between 22:00-02:00 LT. The average EFs of NO, $NO_2$, $NO_x$, and CO before dawn (00:00-05:00 LT) were 2.8, 1.8, 2.1, and 2.5 times, respectively, greater than those during other hours (06:00-23:00 LT) of the day, suggesting that the average EFs of the nighttime fleet were higher than those of the daytime fleet. This could be attributed to the increased proportions of HDV and DVs in the fleet at night (Fig.A3a-b). In the WJL tunnel, the fractions of HDV and DVs in the fleet before dawn (00:00-05:00 LT) were 1.5 and 1.9 times those at 06:00-23:00 LT, respectively. This finding was also consistent

with Shen et al. (2014)'s study in Beijing, which demonstrated that DVs contributed more to emissions at night than during the day. The proportion of HDV was strongly correlated ($R^2$=0.70) with that of DVs because DVs accounted for approximately 87.7% of the total HDV traversing the WJL tunnel.

The diurnal variations in the proportions of HDV (or DVs) in the fleet allowed us to conduct linear regressions (equation (4)). According to the regression results (Fig.3 and Table A2), positive correlations between the average EFs and the proportion

of HDV (or DVs) were observed for $NO$-$NO_2$-$NO_x$, and CO, which further demonstrated the significant contribution of HDV (or DVs) to the before-dawn fleet in Tianjin.

The average EFs for the fleet, GVs/DVs, and LDV/HDV are summarized in Table A2. In general, the average upslope EFs of NO, $NO_2$, and $NO_x$ for HDV were 28, 34, 27 times greater than those for LDV. The average downslope EFs of NO, $NO_x$, and CO for HDV were approximately 37, 23, 55 times greater than those for LDV. Additionally, the average upslope EFs of NO,

$NO_2$, and $NO_x$ for DVs were approximately 29, 25, 27 times greater than those for GVs. The average downslope EFs of NO, $NO_x$, and CO for DVs were approximately 39, 26, 33 times greater than those for GVs. Our results are comparable to those from Dallmann et al. (2013)'s study, which reported that the EFs of organic aerosol and black carbon from heavy-duty diesel trucks were 10 and 50 times higher than those of LDV in the United States in 2010, respectively. However, because of the insignificant variation in the compositions of the vehicular emission standards (from China I to China V) during the measurement campaign

(Fig.A3c), an attempt to differentiate EFs under different emission standards by multiple linear regression (Zhang et al., 2018b; Hwa et al., 2002) failed. Although the EFs estimated by the tunnel tests were considered as EFs in a real-world setting, there were still several limitations, such as the limited speeds and accelerations, and that most of the vehicles were operating under hot-stabilized conditions (Gertler, 2005). During the study campaign, the average proportion of HDV (DVs) traversing the WJL tunnel was 1.56±1.26% (1.97±0.32%). Although the low proportion of HDV (DVs) limited the accuracy of the estimated EFs

of HDV (DVs) using the linear regression model, the fleet-average EFs in the urban tunnel were more representative of urban areas because of the similar fleet composition between the urban tunnel and the urban areas.

### 3.3 Primary vehicular $f_{NO_2}$

The fractions of $NO_2$ in $NO_x$ ($f_{NO_2}$, v/v) from vehicular emissions were determined in this tunnel study. As shown in Fig.4, the ratio of the primary emitted and transformed $NO_2$ to the primary emitted $NO_x$ (Ratio 1) were approximately 0.40±0.21



for downslope (Fig.4a), 0.11±0.23 for upslope (Fig.4e), and 0.19±0.09 for overall (Fig.4f). Excluding the $O_3$ titration effects (Ratio 2), the fleet-average $f_{NO_2}$ by volume were approximately 0.18±0.09 for downslope (Fig.4b), 0.10±0.22 for upslope (Fig.4f), and 0.10±0.05 for overall (Fig.4j). Student's test ($t$-test) indicated that there was a significant ($p$<0.05) decrease for downslope and overall $f_{NO_2}$ from Ratio 1 to Ratio 2 because of the relatively high $O_3$ concentrations at the inlet of the tunnel

(Fig.A1d). However, no significant difference ($p$>0.05) was found between upslope $f_{NO_2}$ in Ratio 1 and Ratio 2 because of the inherent low $O_3$ concentrations from the middle (2#) to the outlet (3#) of the tunnel (Fig.A1d). Considering the primary vehicular $f_{NO_2}$ (Ratio 2), the transformed $NO_2$ excluding Reaction (R1) contributed 22%, 1%, and 9% to Ratio 2 under the downslope, upslope, and overall condition, respectively. Yao et al. (2005)'s tunnel study also estimated the transformed $NO_2$ excluding $O_3$ titration effects contributed 13% to Ratio 2. The upslope primary $f_{NO_2}$ is 44.4% less than those for downslope,

indicating a larger fraction of vehicular NO was emitted during the upslope than during the downslope.

Simmons and Seakins (2012) conducted tunnel measurements in a United Kingdom road tunnel and concluded that the $f_{NO_2}$ showed a pronounced diurnal cycle with an average value of 0.17 (95% confidence intervals: 0.14, 0.18) by volume from 7:00 to 19:00 LT. Our analysis suggested that the downslope $f_{NO_2}$ (Ratio 1 and 2) showed a sharp increase from early morning (06:00 LT) to noon (12:00 LT) and a decrease before dawn (00:00-05:00 LT). The diurnal variation in the downslope $f_{NO_2}$ was

opposite that of the proportion of HDV (Fig.4c) and DVs (Fig.4d), with the proportion being higher before dawn (00:00-05:00 LT) and lower during the day. The increase in downslope $f_{NO_2}$ could be attributable to a decrease in the proportion of HDV (or DVs) in the total fleet because of the negative correlations between the $f_{NO_2}$ and the proportion of HDVs (or DVs), which is consistent with Simmons and Seakins (2012)'s study. However, no obviously diurnal cycles in upslope and overall $f_{NO_2}$ (Ratio 1 and 2) were observed in this study, particularly for the upslope $f_{NO_2}$. Additionally, the correlations between upslope

(or overall) $f_{NO_2}$ and proportions of HDV (or DVs) were statistically insignificant ($p$>0.05). The results suggested that there were no significant differences in upslope $f_{NO_2}$ among different vehicular types (or fuel types).

The primary vehicular $f_{NO_2}$ can vary considerably depending on such parameters as the vehicular type, fuel type, inspection and maintenance system (I/M), vehicle operation conditions (e.g., cold start, idling, low-speed), and after-treatment devices (e.g., selective catalytic reduction systems, diesel particulate filter, diesel oxidation catalyst) (O'Driscoll et al., 2016; Williams

and Carslaw, 2011; Shon et al., 2011; Alvarez et al., 2008; Soltic and Weilenmann, 2003). O'Driscoll et al. (2016) reported that the average $f_{NO_2}$ was 0.44±0.20, and the selective catalytic reduction (SCR) systems had a higher $f_{NO_2}$ (0.55±0.12) for Euro VI diesel passenger cars based on the PEMS. The future $f_{NO_2}$ was projected to increase with the introduction of diesel-oxidation-catalysts (DOC), other after-treatment devices for DVs, and the improvement of fuel quality to meet the successively stringent emission standards in China.

## 3.4   Emission factors of NMVOCs

The average EFs for the 99-target NMVOC species measured in the tunnel in Tianjin are shown in Table A4 (zero emission factors for some NMVOCs results from $C_{Outlet}$ less than or equal to $C_{Inlet}$). The EFs for NMVOCs reported from a tunnel study in Hong Kong in 2003 (Ho et al., 2009), Guangzhou in 2014 (Zhang et al., 2018c), Nanjing in 2015 (Zhang et al., 2018b), and Hong Kong in 2015 (Cui et al., 2018) are also listed in Table A4. The fleet-average EFs (mg km$^{-1}$ veh$^{-1}$) for the NMVOCs





in the tunnel were 40.56±12.18, which was lower than those of previous studies in Hong Kong in 2003 (568 mg km$^{-1}$ veh$^{-1}$) (Ho et al., 2009), Guangzhou in 2014 (1.10×10$^3$ mg km$^{-1}$ veh$^{-1}$) (Zhang et al., 2018c), Nanjing in 2015 (373.88 mg km$^{-1}$ veh$^{-1}$) (Zhang et al., 2018b), and Hong Kong in 2015 (58.8±50.7 mg km$^{-1}$ veh$^{-1}$) (Cui et al., 2018). This might be attributed to the improvement of fuel quality and emission standards in China. By group, alkanes, alkenes, aromatics, halocarbons, and

5 OVOCs had average EFs of 19.59±6.84, 5.52±1.31, 9.02±2.25, 0.20±0.14, and 5.29±2.22 mg km$^{-1}$ veh$^{-1}$, respectively (Fig.5a).

 The five most abundant VOC species (Fig.6) in the vehicular emissions were, in decreasing order, isopentane (6.80 mg km$^{-1}$ veh$^{-1}$), toluene (3.29 mg km$^{-1}$ veh$^{-1}$), ethylene (2.56 mg km$^{-1}$ veh$^{-1}$), methyl tert-butyl ether (MTBE: 2.48 mg km$^{-1}$ veh$^{-1}$), and n-pentane (2.14 mg km$^{-1}$ veh$^{-1}$) (Table A4), accounting for 37.90±5.37% of the total NMVOCs emissions. The

10 abundance (v/v) of isopentane (12.89±5.98%), toluene (8.01±2.51%) and n-pentane (4.88±1.41%) were high in the emission of the tunnel fleet (Fig.6), which is consistent with Tsai et al. (2006)'s study. These three species are primary indicators of gasoline evaporation (Liu et al., 2008; Yue et al., 2017; Hwa et al., 2002; Tsai et al., 2006), implying that the control of vehicular evaporative emission becomes increasingly important with the improvement in fuel quality and emission standards in China (Liu et al., 2015a). It should be noted that MTBE, a gasoline oxygenation additive used to increase the octane number

15 and reduce vehicular emissions of carbon monoxide, accounted for 5.73±1.53% of the vehicular NMVOCs emissions. MTBE is a maker specific to gasoline-related sources and it can be present in both gasoline vapors and gasoline-powered vehicle exhausts due to incomplete combustion (Zhang et al., 2013; Poulopoulos and Philippopoulos, 2000).

 Our previous tunnel study in Nanjing in 2015 (Zhang et al., 2018b) reported that the five most abundant VOC species in vehicle emissions were, in decreasing order, ethane (52.47±6.72 mg km$^{-1}$ veh$^{-1}$), isopentane (17.82±11.97 mg km$^{-1}$ veh$^{-1}$),

20 propane (11.80±3.48 mg km$^{-1}$ veh$^{-1}$), ethylene (10.17±1.05 mg km$^{-1}$ veh$^{-1}$), and toluene (9.36±5.27 mg km$^{-1}$ veh$^{-1}$). The relatively high EFs of ethane and propane measured in Nanjing in 2015 are associated with the large fractions of liquefied natural gas (LNG) fueled vehicles (13%) in the fleet. The most abundant VOC species in vehicular emissions measured in a tunnel in Hong Kong in 2003 (Ho et al., 2009) was ethylene (12.6±4.3 mg km$^{-1}$ veh$^{-1}$), followed by toluene (12.1±3.9 mg km$^{-1}$ veh$^{-1}$), n-butane (8.7±3.1 mg km$^{-1}$ veh$^{-1}$), propane (5.7±2.5 mg km$^{-1}$ veh$^{-1}$), and isopentane (5.6±2.1 mg km$^{-1}$

25 veh$^{-1}$). Generally, isopentane, toluene, and ethylene are the three species frequently observed as the most abundant VOC species in vehicular emissions.

 NMVOCs are precursors of O$_3$ formation. The EFs of the O$_3$ formation potential for each VOCs were estimated by the MIR coefficient times the EF for each NMVOC species. The total O$_3$ formation potential was approximately 135.14±36.78 mg km$^{-1}$ veh$^{-1}$, which is lower than those reported from previous studies in Hong Kong in 2003 (568 mg km$^{-1}$ veh$^{-1}$)

30 (Ho et al., 2009), Guangzhou in 2014 (1.10×10$^3$ mg km$^{-1}$ veh$^{-1}$) (Zhang et al., 2018c), and Nanjing in 2015 (373.88 mg km$^{-1}$ veh$^{-1}$) (Zhang et al., 2018b). The lower OFP observed in this study is likely attributed to the progression of vehicular technology combined with emission standards (China IV: 40.20±0.85%, China V: 47.76±1.29%). As shown in Fig.5b, the OFPs of alkanes, alkenes, aromatics, halocarbons, and OVOCs were 23.59±9.05, 49.25±11.78, 42.22±14.36, 0.04±0.03, and 15.92±7.77 mg km$^{-1}$ veh$^{-1}$, respectively. The alkenes (aromatics) accounted for 13.90±2.80% (22.8±4.9%) of total




vehicular NMVOCs emissions observed in the tunnel but were nevertheless responsible for 37.60±6.41% (32.2±5.77%) of the total vehicular OFP.

The top five VOC species as the largest contributors to O$_3$ production for individual vehicles were ethylene (22.75±4.24 mg km$^{-1}$ veh$^{-1}$), toluene (12.94±4.36 mg km$^{-1}$ veh$^{-1}$), $m/p$-xylene (12.89±5.47 mg km$^{-1}$ veh$^{-1}$), propylene (10.99±4.00 mg km$^{-1}$ veh$^{-1}$), and isopentane (7.93±4.47 mg km$^{-1}$ veh$^{-1}$) (Fig.6). These findings were consistent with our previous tunnel study in Nanjing in 2015 (Zhang et al., 2018b). Ethylene emissions contributed 17.46±3.84% to the measured VOCs reactivity for vehicular emissions, which was nearly 20.12% in Nanjing in 2015 (Zhang et al., 2018b) and 23% in Hong Kong in 2003 (Ho et al., 2009).

### 3.5  Source identification of NMVOCs

The fleet-average source profiles of NMVOCs were estimated in section 3.4 by the emission factor method (equation (1)). The VOCs from vehicular emissions were mainly comprised of tailpipe and evaporative emissions. Thus, the fleet-average source profiles were mixed by two factors, tailpipe emissions and evaporative emissions. In the present work, we attempted to distinguish source profiles of NMVOCs from tailpipe and evaporative emissions by PMF methods.

For each NMVOC species, the corresponding inlet concentrations were subtracted (ΔNMVOCs: $C_{Outlet} - C_{Inlet}$) to isolate the emission signals from vehicles traveling through the tunnel. To minimize the uncertainties caused by the NMVOC species with low ΔNMVOCs values, the top 30 NMVOC species (Fig.6) from vehicular emissions were used in source apportionment by PMF. The highly reactive NMVOCs species were not excluded in the source apportionment because of the low chemical reactions in the tunnel where there is weaker solar radiation, which is different from traditional source apportionment of ambient VOCs (Zheng et al., 2018; Liu et al., 2016).

Two factors (Factor1 and Factor2) were resolved by the PMF analysis and their contribution to each species (Fig.7a) and source profiles (Fig.7b-c) are shown. Approximately 97.07% of the measured NMVOCs from vehicular emissions were explained using the PMF (Fig.A4a). Moreover, for the top 30 individual NMVOCs species, the PMF model also reproduced the predicted values well, with the average R$^2$ being 0.71±0.24 (Table.A3). Therefore, we considered that the ΔNMVOCs concentrations in the tunnel could be resolved by the two factors using the PMF model.

The fleet-average EFs of isopentane, toluene, n-pentane, and n-butane generally showed good correlations with ambient temperature (Fig.8), suggesting that these four compounds could be recognized as primary indicators of evaporation emissions (Yue et al., 2017; Liu et al., 2008; Tsai et al., 2006; Hwa et al., 2002). High loadings of isopentane (65.53%), toluene (54.87%), n-pentane (70.29%), and n-butane (73.18%) were found in Factor1. The source apportionment of the NMVOC samples at 3-h intervals provide a unique opportunity to discuss the diurnal variations in factor contributions (Fig.A4b). Factor1 had a higher contribution to vehicular VOC emissions during the daytime, and a lower contribution during the nighttime. In addition, diurnal variations in Factor1 contribution linearly correlated with diurnal variations in temperatures ($y = 0.0755x(°C) - 1.8173, R^2 = 0.722$). Therefore, Factor1 was identified as evaporative emissions, and Factor2 was identified as tailpipe emissions.

As listed in Table.A5, the EFs of the top 30 VOC compounds in vehicular evaporative and tailpipe emissions were estimated via relative contributions (Fig.7a) and fleet-average EFs (Fig.A4). During the measurement campaign, evaporative





emissions accounted for nearly half (49.33±22.90%) the total vehicular NMVOC emissions. However, the tailpipe emissions of NMVOCs might be underestimated due to the downslope (1#-2#) road conditions. A previous tunnel study noted that the upslope EF of NMVOCs is only 1.3 times that of the downslope (Chang et al., 2008). Thus, the upslope evaporative emissions were estimated to account for approximately 42.82% of the total vehicular upslope NMVOC emissions, supposing that there

was no significant difference between evaporative emissions under downslope and upslope conditions, and the upslope EF of NMVOCs is 1.3 times that of the downslope. It should be noted that the evaporative NMVOC emissions estimated from tunnel tests are mostly running loss emissions, other evaporative emissions (i.e., diurnal, hot soak, permeation, and refueling process) were generally not included, which are also major components of vehicular VOCs emissions. Thus, the evaporative and tailpipe emissions contribute equally to NMVOC emission inventories, air quality, and energy. Nevertheless, the relative

contributions of evaporative NMVOC emissions to total vehicular NMVOC emissions are found to be temperature-dependent with increasing ratio of 7.55% $°C^{-1}$ in this study (average ambient temperature: 27.13±2.32), suggesting the contribution of evaporative NMVOC emissions estimated during the study period might be overestimated in terms of the whole year.

## 3.6  BTEX and diagnostic ratios

The aromatic group of NMVOCs, in particular benzene, toluene, ethylbenzene, $p$-xylene, $m$-xylene and $o$-xylene, which are

toxic VOCs known as the BTEX group and have been the subject of intensive studies (Zhang et al., 2018b, 2016a). Emissions from combustion of gasoline and diesel fuels are the largest contributions to ambient BTEX concentrations (Bolden et al., 2015). According to our tunnel tests in Tianjin, the EFs (mg $km^{-1}$ $veh^{-1}$) were 1.97±0.52 for benzene, 0.66±0.26 for ethylbenzene, 3.29±1.10 for toluene, 0.48±0.19 for $o$-xylene, and 1.42±0.42 for $m/p$-xylene. The emission level of BTEX in this study was compared to that of reported tunnel studies in Taiwan in 2000 (Hwa et al., 2002), Hong Kong in 2003 (Ho et al.,

2009), Guangzhou in 2004 (Zhang et al., 2018c), Taiwan in 2005 (Hung-Lung et al., 2007), Guangzhou in 2014 (Zhang et al., 2018c), Nanjing in 2015 (Zhang et al., 2018b), and Hong Kong in 2015 (Cui et al., 2018). As shown in Fig.9, the BTEX levels tested in recent four years (Guangzhou in 2014, Nanjing in 2015, Hong Kong in 2015, and Tianjin in 2017) decreased when compared to those tested during the last decades (Taiwan in 2000, Hong Kong in 2003, Guangzhou in 2004, and Taiwan in 2005). Cui et al. (2018) and Zhang et al. (2018c) noted that the EFs of VOCs from on-road vehicles decreased by 44.7% in

Hong Kong from 2003 to 2015, and 9% in Guangzhou from 2004 to 2014, respectively. We conducted the linear regression between BTEX levels and the test year from previous studies and this study, and found the average BTEX levels in China have dramatically decreased during the last decade.

Since the 1990s, China has started vehicular emission control programs, including controls for new vehicles and in-use vehicles, improvements in fuel quality (e.g., the promotion of unleaded gasoline and low-sulfur fuels), promotion of alternative

fuels and incentives for electric vehicles (Wu et al., 2017). Emission standards have been upgraded from China I (equals to Euro I) in 2001 to China V in 2015. By estimation from the regressions, the EFs (mg $km^{-1}$ $veh^{-1}$) in 2000 (before China I) were 12 (95% confidence interval (CI): 5.04, 19.14) for benzene, 5.87 (95%CI: 1.82, 9.92) for ethylbenzene, 29.4 (95%CI: 18.68, 39.97) for toluene, 7.3 (95%CI: 3.53, 11.06) for $o$-xylene, and 12.41 (95%CI: 0.86, 23.96) for $m/p$-xylene. The EFs (mg $km^{-1}$ $veh^{-1}$) in 2017 (after China V) were 1.89 (95%CI: 0, 8.31) for benzene, 2.33 (95%CI: 0, 6.02) for ethylbenzene,




5.24 (95%CI: 0, 14.93) for toluene, 0.92 (95%CI: 0, 4.35) for $o$-xylene, and 3.54 (95%CI: 0, 14.04) for $m/p$-xylene. The EFs from on-road vehicles decreased by 84% for benzene, 60% for ethylbenzene, 82% for toluene, 87% for $o$-xylene, and 71% for $m/p$-xylene. Generally, the BTEX levels decreased by approximately 79% when emission standards increased from China I to China V. Our result demonstrate that an improvement in fuel quality can significantly reduce vehicular BTEX emissions and

bring great health benefits for human beings.

Table 2 presents ratios (v/v) of paired VOCs that have been widely used to indicate relative contributions from different emission sources (Zhang et al., 2013). Among the different NMVOCs measured in the tunnel, toluene and benzene are of particular interest. The ratio of these two NMVOCs is often used either to discriminate traffic emission sources from other anthropogenic sources (Ait-Helal et al., 2015; Steinbacher et al., 2005; Hedberg et al., 2002) or as a chemical clock for the

determination of the photochemical age of air masses (Nelson and Quigley, 1983). The average toluene-to-benzene ratio (T/B) were 0.78±0.17 for the inlet, and 1.00±0.14 for the middle of the tunnel. Additionally, the vehicular T/B ratio was 1.42±0.33. The T/B ratios generally ranged from 2.0-5.8 in refueling vapors (Zhang et al., 2013), 1.6-5.6 in headspace vapors (Zhang et al., 2013; Liu et al., 2008; Na et al., 2004; Harley et al., 2000), and 1.42-4.19 in tunnel studies (Table 2). Gelencsér et al. (1997) noted that T/B increased as the measurement site moved closer to the vehicular emission sources. In our case, the average T/B

measured at the middle was larger than that measured at the inlet of the tunnel, further validating T/B concentration ratio as a tool for characterizing the contributions of vehicular emission sources. The T/B ratio for evaporative and tailpipe emissions was 1.56 and 1.23 (v/v), respectively. A higher ratio of the toluene and benzene from evaporation was observed because of the high toluene concentration in gasoline in Asia (Liu et al., 2015a; Zhang et al., 2013).

The ratios of $i$-butane/n-butanes ($i$-B/n-B) ranged from 0.28 (this study) to 0.94 (previous study in Nanjing). A high $i$-B/n-B

ratio was associated with a high proportion of LNG fueled vehicles (Russo et al., 2010). Our previous tunnel study in Nanjing reported that the proportion of LNG vehicles in Nanjing were 13% in the fleet. In this study, the alternative-fueled (including LNG) vehicles only accounted for 3.8±0.32% in the fleet in Tianjin. The $i$-pentane/n-pentane ($i$-P/n-P) ratios ranged from 1.31 to 3.64 according to the tunnel tests. The MTBE/Benzene (MTBE/B) and MTBE/Toluene (MTBE/T) ratios ranged from 0.38-1.12 and 0.18-0.79, respectively. The MTBE/B and MTBE/T ratios measured in the tunnel tests were much lower than

that of 4.1-6.6 and 0.7-3.3, respectively, in refueling vapors reported by Zhang et al. (2013). Higher MTBE/B and MTBE/T ratios in gasoline vapors might be indicators to differentiate between gasoline evaporative and tailpipe emissions. The vehicular exhausts tested in tunnel tests had ratios of B, T, E, $m,p$-X, and $o$-X to 2,2-DimB ranged from 2.87-24.82, 12.00-56.12, 1.90-8.93, 0.54-5.40, and 2.28-6.57, respectively, significantly higher than those of 0.4-1.5, 0.9-4.8, 0.03-1.0, 0.1-1.3, and 0.04-0.8, in gasoline and headspace vapors (Zhang et al., 2013; Na et al., 2004; Harley et al., 2000). Thus, MTBE/B and BTBE/T ratios,

and ratios of B, T, E, $m,p$-X, and $o$-X to 2,2-DimB, were likely able to differentiate vehicular sources from mixed emission sources.

### 3.7   The relationship between NMVOCs and NO$_x$

The average NMVOCs/NO$_x$ ratio was approximately 3.43±1.50 ppbC/ppbv at the inlet and 2.74±0.90 ppbC/ppbv at the middle of the tunnel, much lower than the 8:1 threshold. This indicates that the O$_3$ formation was likely to be VOCs-limited





at near-road sites. The diurnal variations in the NMVOCs/NO$_x$ ratios for the middle of the tunnel showed a similar trend to those for the inlet, with a V-shape being minimum during the day (09:00-18:00 LT) and maximum at night (18:00-09:00 LT) (Fig.10a). However, the diurnal variation in the NMVOCs/NO$_x$ ratios measured in the ambient air generally showed a signal peak tendency with a maximum appearing at 14:00-15:00 LT (Liu et al., 2016; Zou et al., 2015). This discrepancy might be

associated with the photochemical reactions in the ambient air. In the absence of solar irradiation at the inlet and outlet of the tunnel, O$_3$-involved photochemical reactions were generally inhibited.

Fig.10b shows the linear relationships between $\Delta$NMVOCs (i.e., $C_{Outlet} - C_{Inlet}$) and $\Delta$NO$_x$, and $\Delta$NO$_x$ accounted for 80%, 80%, and 94% of the total variance in the NMVOCs expressed as a concentration by volume (ppbv), carbon atoms (ppbC), and mass ($\mu$ g m$^{-3}$), respectively. The good correlations between $\Delta$NMVOCs and $\Delta$NO$_x$ indicated that there were

relatively stable average ERs of the NMVOCs to NO$_x$ for on-road vehicles. Based on the slope of the regression line, the ER of the NMVOCs to NO$_x$ during the measurement campaign was approximately 0.84 (95%CI: 0.67, 1.01) for VOCs expressed as the concentration of carbon atoms (ppbC), much lower than the 8:1 threshold. Thus, O$_3$ chemical regimes were more likely to be VOCs-limited for cities or regions where VOCs and NO$_x$ emissions are dominated by vehicular exhaust. Liu et al. (2016) and Han et al. (2015) noted that vehicular activities were the most significant contributor to the ambient VOCs in Tianjin. In

addition, Wu et al. (2016) and Wang et al. (2016) also found that vehicular emission was the highest contributor to VOCs in Guangzhou and Beijing, respectively. O$_3$ chemical regimes were generally observed to be VOCs-limited in major Chinese cities (Xue et al., 2014), which might be attributable to the low ER of the NMVOCs to NO$_x$ from on-road vehicles. Moreover, good correlation (R$^2$=0.94) between $\Delta$NMVOCs and $\Delta$NO$_x$ by mass concentration were also observed in our study, indicating the on-road vehicular emission of NMVOCs was proportional to the NO$_x$, and the ER was approximately 2.04 (95%CI: 1.83, 2.26).

It should be noted that the ER values should be carefully treated to evaluate the NMVOCs emissions from NO$_x$ emissions, because of the differences between real-world vehicles, including vehicular technology, fleet composition, driving conditions, and other variables.

To understand the interannual trends of the vehicular emission ratios of VOCs/NO$_x$, a critical value in identifying O$_3$ chemical regime, we estimated vehicular emissions of NO$_x$, CO, and VOCs in Tianjin from 2000 to 2016 (Fig.11a-c). The

COPERT IV model, an "average speed" model, was used in this study to estimate vehicular emissions in Tianjin because Chinese vehicular emission standards are similar to Europe's (Jing et al., 2016; He et al., 2016; Sun et al., 2016). Total emissions of NO$_x$ from the Tianjin vehicle fleet increased by 15.17% from 2000 (60.19 Gg) to 2011 (69.32 Gg), and decreased by 20.46% from 2011 to 2016 (55.14 Gg). Heavy-duty trucks (HDT) were the largest contributors of vehicular NO$_x$ emissions since 2012 (Song et al., 2018), and their contribution has increased from 35.47% in 2000 to 46.86% in 2016. The Total emissions of

CO and VOCs from the Tianjin vehicular fleet decreased by 52.47% and 51.44%, respectively. The results suggested that the environmental benefits from the decreased vehicular EFs of CO and VOCs from 2000 to 2016 (Fig.9) were great enough to offset the increased vehicle-kilometers traveled (VKT) and larger fleets. Passenger cars (PC) were the primary source of vehicular CO and VOCs emissions and accounted for 44.11% and 46.14% in 2000 to 77.42% and 81.75% in 2016.

We estimated annual emission ratios of VOCs/NO$_x$ in Tianjin from 2000 to 2016 (Fig.11d), and in China from 2010 to

2030 (Fig.11e). The estimated total vehicular emission ratios of VOCs to NO$_x$ in China for NAP (Standards are assumed at



the same level of 2013 during the future years (e.g., China IV for LDGVs and China III for HDDVs)), PC1 (increasingly stringent emission standards and improved fuel quality are implemented according to a probably timetable), and PC2 scenarios (for internal combustion engine vehicles, the implementation of future emission standards is consistent with that under the PC1 scenario) from 2010 to 2030 were based on Wu et al. (2017)'s study. The annual emission ratios of VOCs/NO$_x$ from

the Tianjin vehicle fleet decreased from 1.16 in 2000 to 0.62 in 2016. We estimated the emission ratio of VOCs/NO$_x$ to be approximately 2.04 (95%CI: 1.83, 2.26) through the tunnel test (Fig.10b), which is almost 3.3 times greater than that estimated from the COPERT IV model. This discrepancy might be attributable to the differences in fleet compositions between the tested tunnels and cities, especially cities that HDT are often not allowed to access to (Song et al., 2018). In general, PC (3.01±0.17) and LDV (3.07±1.47) have higher emission ratios of VOCs/NO$_x$ than HDT (0.15±0.04) and BUS (0.17±0.07) from 2000 to

2016. The high proportion of HDT (1.11%) in the Tianjin vehicular fleet, 5 times greater than that in the tunnel fleet (0.16%), led to lower emission ratios of VOCs/NO$_x$ than those in the tunnel test. The vehicular emission ratios of VOCs/NO$_x$ in China from 2010 to 2016 have a good agreement ($r$=0.84, $p$<0.05) with those in Tianjin. The emission ratios of VOCs/NO$_x$ in PC1 and PC2 scenarios were projected to increase from 0.45 and 0.40 in 2017 to 0.80 and 0.71 in 2030, respectively. And the emission ratios of VOCs/NO$_x$ under current emission standards were to remain nearly unchanged (0.33) since 2020. As even

more stringent emission standards are properly implemented in the future, the O$_3$ chemical regimes were more likely to remain VOCs-limited (i.e., 8:1 threshold) for cities or regions where VOCs and NO$_x$ emissions are dominated by vehicular exhaust.

## 4   Conclusions

Based on a more-than-two-week test of on-road vehicular VOCs-NO$_x$-CO emissions in the Wujinglu tunnel in urban Tianjin in northern China, and on the use of the statistical regression model, PMF receptor model, and COPERT IV model, this study

allowed for characterizing the primary VOCs-NO$_x$-CO emissions from vehicular exhaust systematically. The fleet-average EFs of VOCs, NO-NO$_2$-NO$_x$, and CO under upslope, downslope and overall road conditions were measured through tunnel tests. The ratios (v/v) of NO$_2$ to NO$_x$ in the primary vehicular exhaust were approximately 0.18±0.09, 0.10±0.22 and 0.10±0.05 for downslope, upslope, and the entire tunnel, respectively. The fleet-average EFs for the 99-target NMVOCs in the tunnel were 40.56±12.18 mg km$^{-1}$ veh$^{-1}$, which was lower than the previous studies. This study found that the improvement of

fuel quality (from China I to China V) can significantly reduce feet-average EFs of VOCs (especially for BTEX). The BTEX (benzene, toluene, ethylbenzene, $p$-xylene, $m$-xylene and $o$-xylene) levels decreased by approximately 79% when emission standards increased from China I to China V. Evaporative (mostly running loss) and tailpipe VOC emissions in tunnel tests could be resolved by the PMF model, and the evaporative and tailpipe emissions contributed equally to NMVOC emission inventories, air quality, and energy. The relative contributions of evaporative NMVOC emissions to total vehicular NMVOC

emissions are found to be temperature-dependent with increasing ratio of 7.55% °C$^{-1}$. Our study enriched the database on the fleet-average emission factors of on-road vehicles for emission inventory, air quality modeling, and health effects studies, provided implications for following O$_3$ control in China from the view of primary emission, and highlighted the importance of further control of evaporative emissions.



*Competing interests.* The authors declare that they have no conflict of interests.

*Acknowledgements.* We thank the members of Center for Urban Transport Emission Research (CUTER) in Nankai University for their contributions to the field study. The authors are also very grateful to the management of the Wujinglu Tunnel for support during sample collection. This work is funded by the National key research and development program of China (2017YFC0212104) and the National
5  Natural Science Foundation of China (21607081). The authors declare they have no completing financial interests.



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





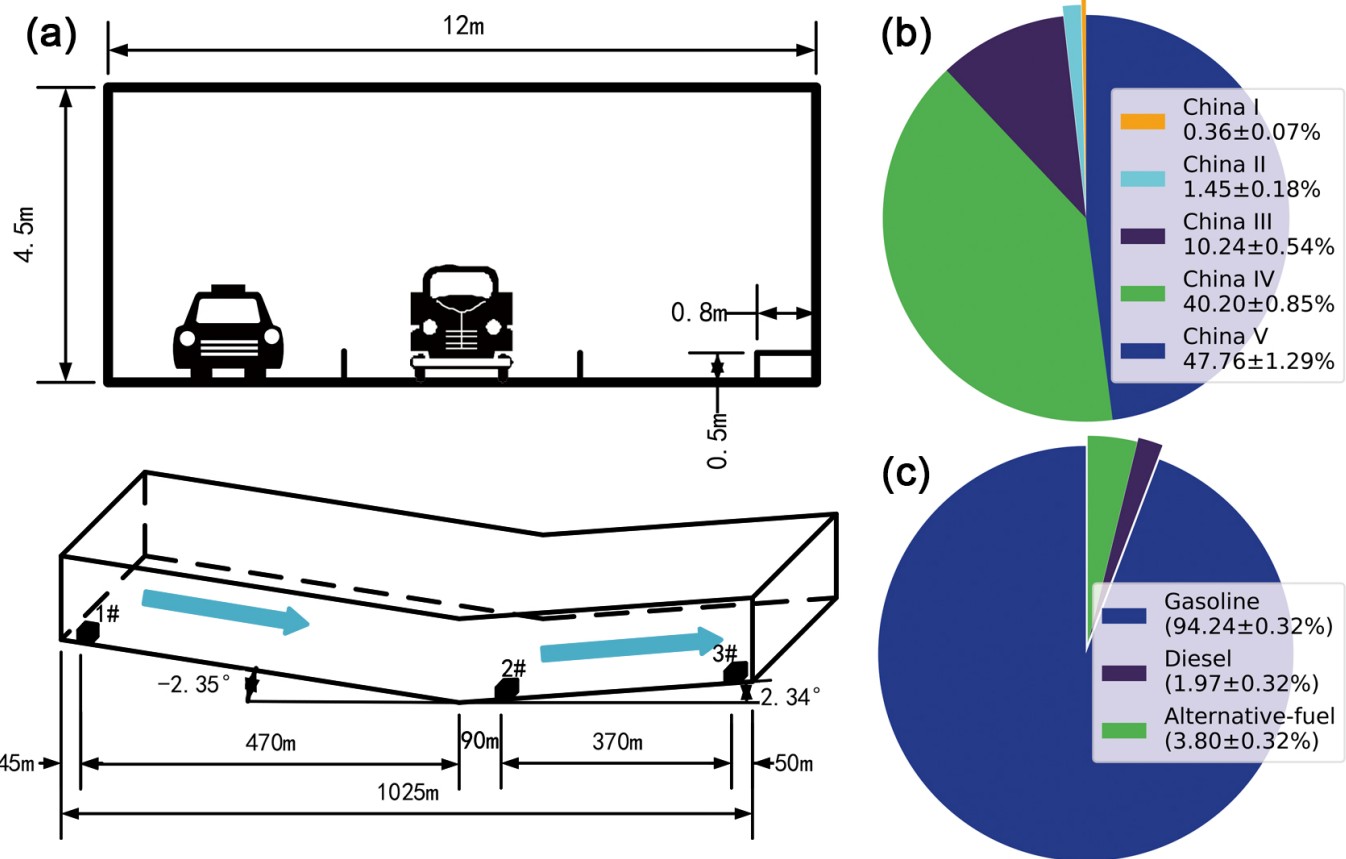

**Figure 1.** The description of (a) WJL tunnel, and the fleet compostion in (b) vehicle types and (c) in-use fuel types.



**Figure 2.** The diurnal variations (represented by boxplots) of emission factors of (a-c) NO, (d-f) $NO_2$, (g-i) $NO_x$, and (j-l) CO for on-road vehicles under downslope (1#-2#), upslope (2#-3#) and overall (1#-3#) road conditions. The average emission factors (mean±standard deviation) were given on the upper right corner of each subfigure.

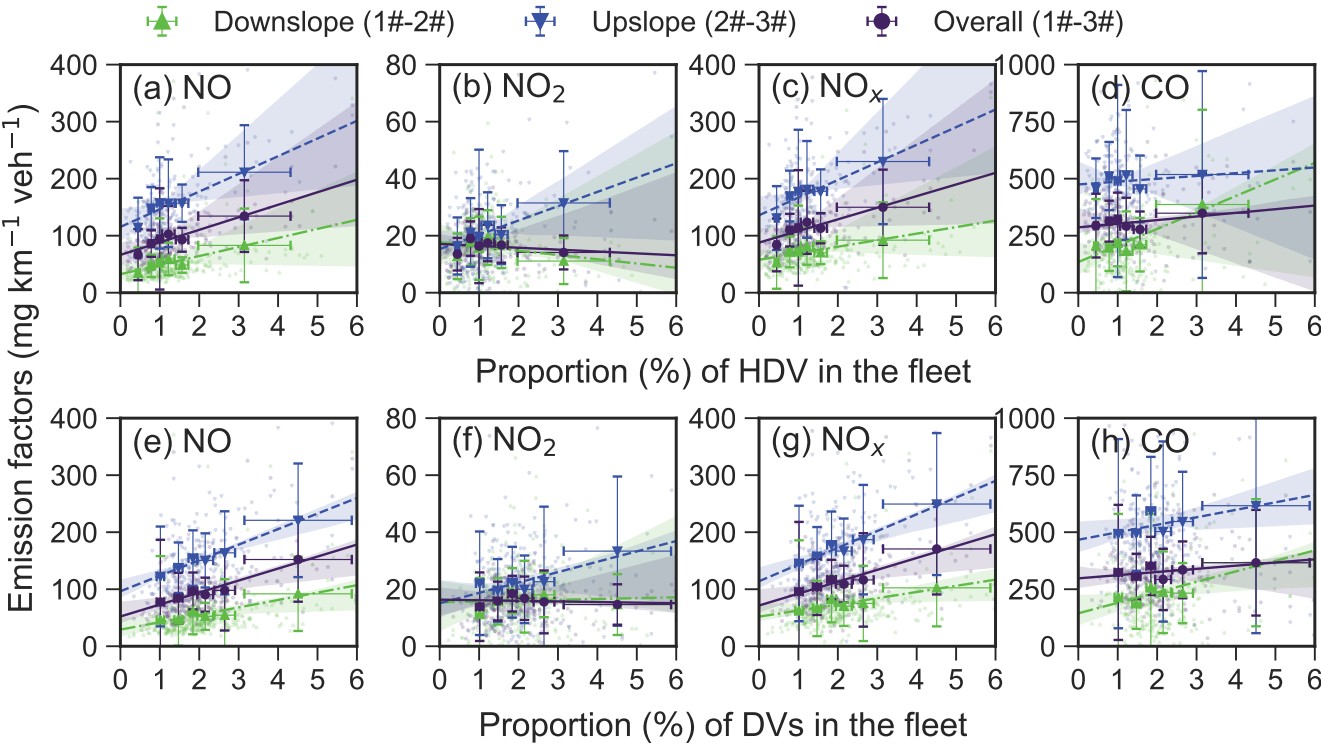

**Figure 3.** The linear regressions (dash-dot regression lines for the downslope, dash-dash regression lines for the upslope, and solid regression lines for the overall condition) between fleet-average emission factors and proportion of heavy-duty vehicles (HDV, a: NO; b: $NO_2$; c: $NO_x$; d: CO) and diesel vehicles (DVs, e: NO; f: $NO_2$; g: $NO_x$; h: CO). The parameters of the linear regressions were listed in Table A2.





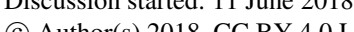

**Figure 4.** The diurnal variations (represented by boxplots) of $f_{NO_2}$ (Ratio 1: a-e-i, Ratio 2: b-f-j), and the relationships between $f_{NO_2}$ and the proportion of heavy-duty vehicles (pHDV: c-g-k) and diesel vehicles (pDVs: d-h-l). The parameters of the linear regressions (dash-dot regression lines for Ratio 1, solid regression lines for Ratio 2) were listed in Table A2.





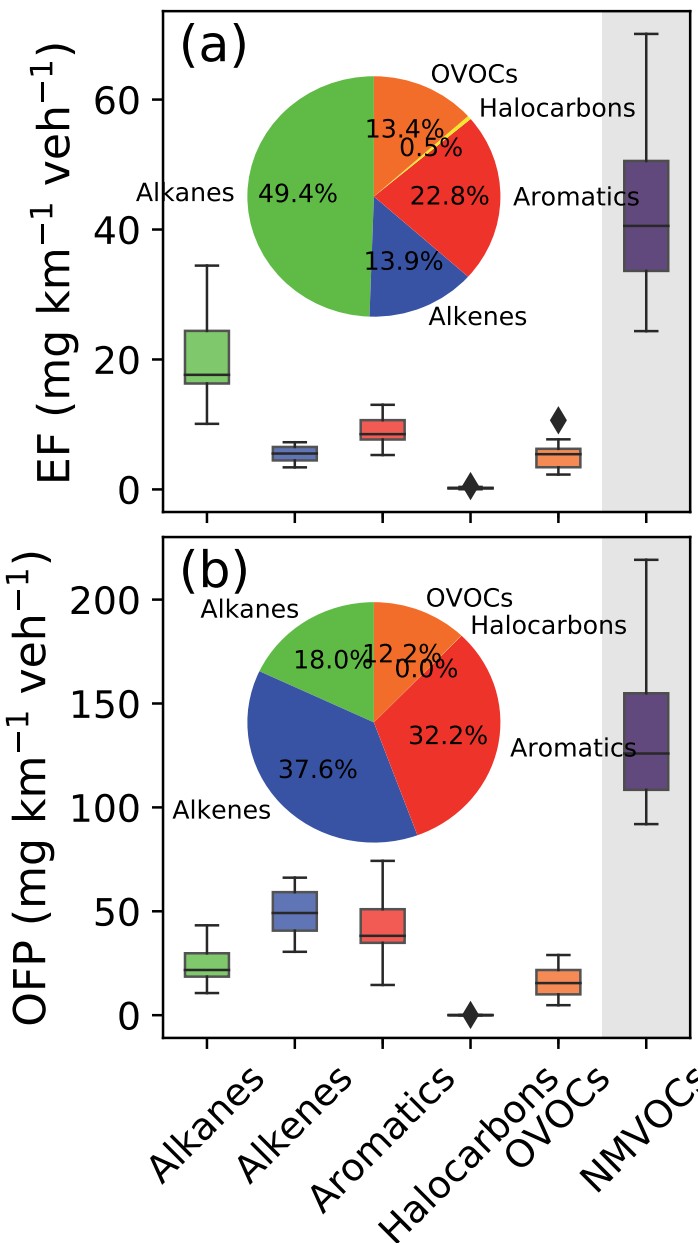

**Figure 5.** (a) The boxplots of fleet-average emission factors (EFs) for the VOC groups, and (b) the EFs of the ozone formation potential (OFP) for the VOC groups.





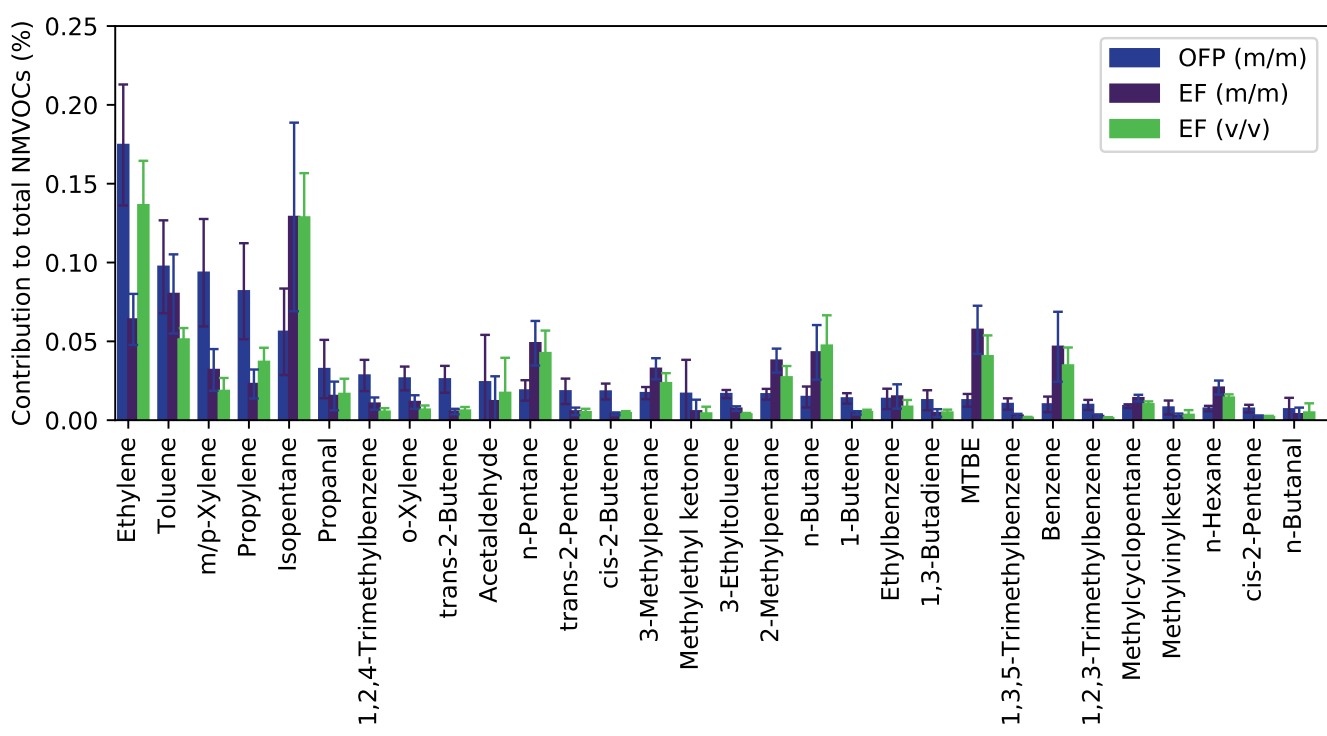

**Figure 6.** The fleet-average OFP (m/m), EF(m/m) and EF (v/v) of top 30 NMVOC species from vehicle emissions (sorted by OFP).





**Figure 7.** (a) The relative factor contributions to individual VOC species; (b) the source profiles of VOCs by volume; (c) the source profiles of VOCs by volume percentage.





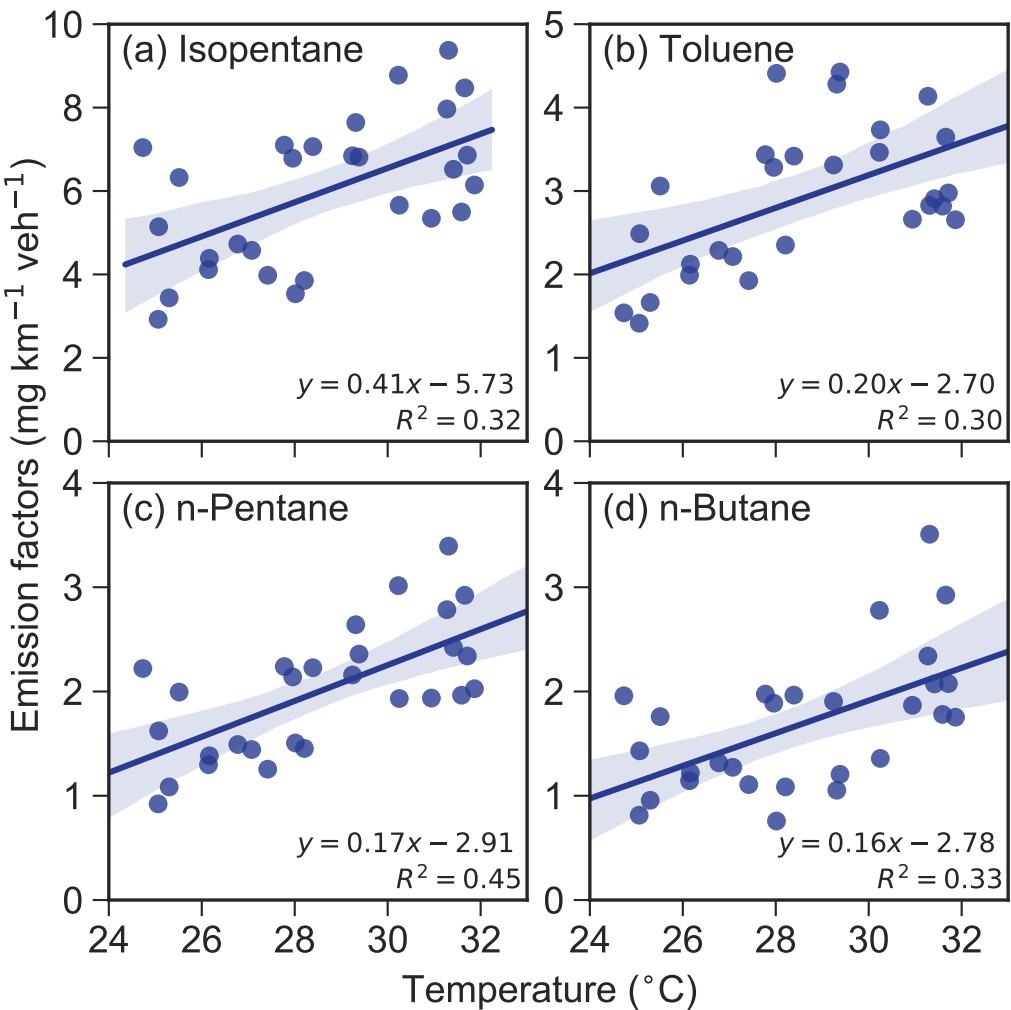

**Figure 8.** The linear regressions (filled with 95% regression confidence intervals) between emission factors of (a) Isopentane, (b) Toluene, (c) n-Pentane, and (d) n-Butane against ambient temperature.



**Figure 9.** The BTEX levels observed in different tunnel studies (Taiwan in 2000 (Hwa et al., 2002)l; Hong Kong in 2003 (Ho et al., 2009);

Guangzhou in 2004 (Zhang et al., 2018c); Taiwan in 2005 (Hung-Lung et al., 2007); Guangzhou in 2014 (Zhang et al., 2018c); Nanjing

in 2015 (Zhang et al., 2018b); Hong Kong in 2015 (Cui et al., 2018)), and the linear regressions (dash-dot lines filled with 95% regression

confidence intervals) between the emission factors of BTEX and the test years.



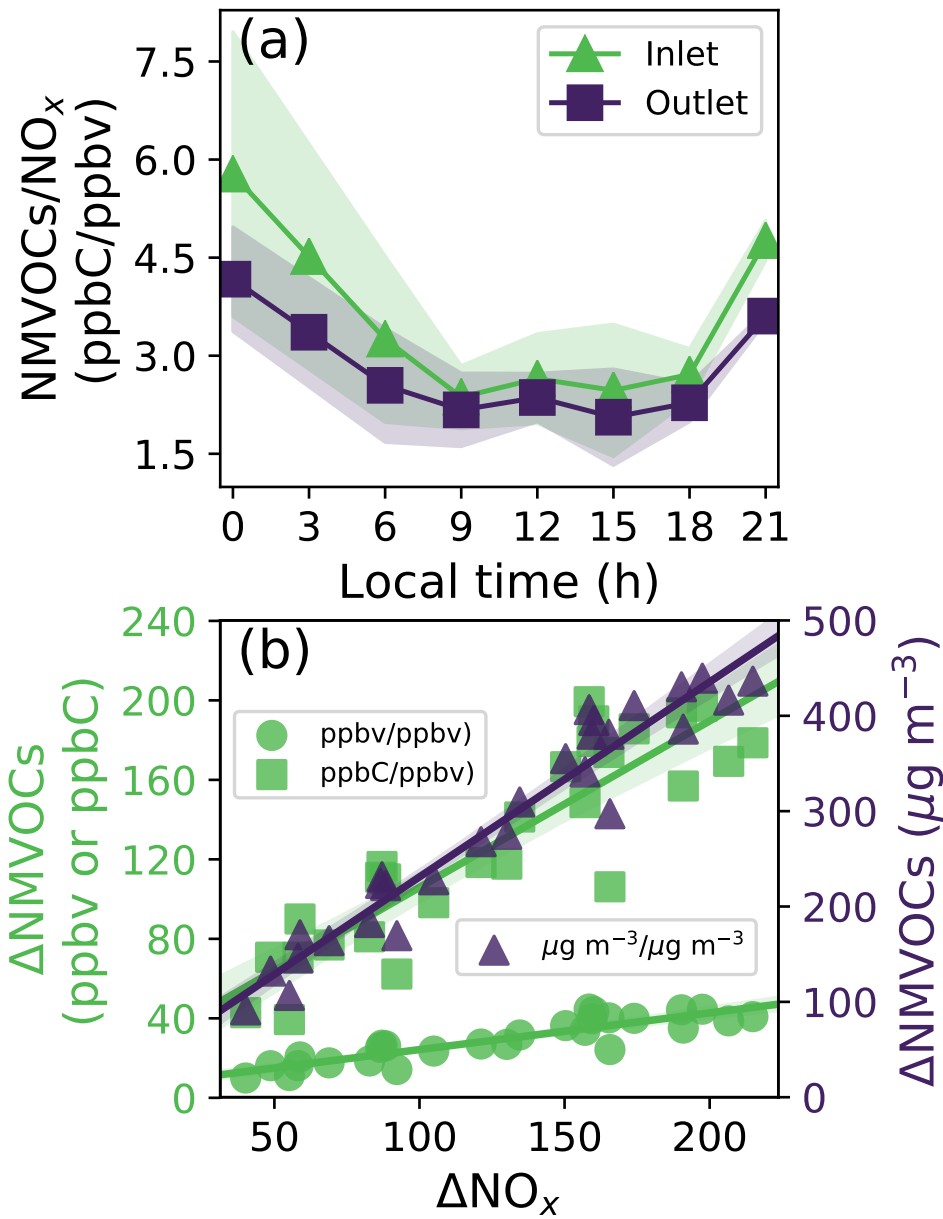

**Figure 10.** (a)The diurnal variations (filled with one standard deviation) of the NMVOCs/NO$_x$ in the inlet and outlet of the tunnel, and (b) the linear regression (filled with 95% confidence intervals) between ΔNMVOCs (i.e., $C_{Outlet} - C_{Inlet}$) and ΔNO$_x$ (i.e., $C_{Outlet} - C_{Inlet}$), ΔNMVOCs (ppbv)=0.18×ΔNO$_x$ (ppbv)+5.80, ΔNMVOCs (ppbC)=0.84×ΔNO$_x$ (ppbv)+22.16, ΔNMVOCs ($\mu$g m$^{-3}$)=2.04×ΔNO$_x$ ($\mu$g m$^{-3}$)+26.59.





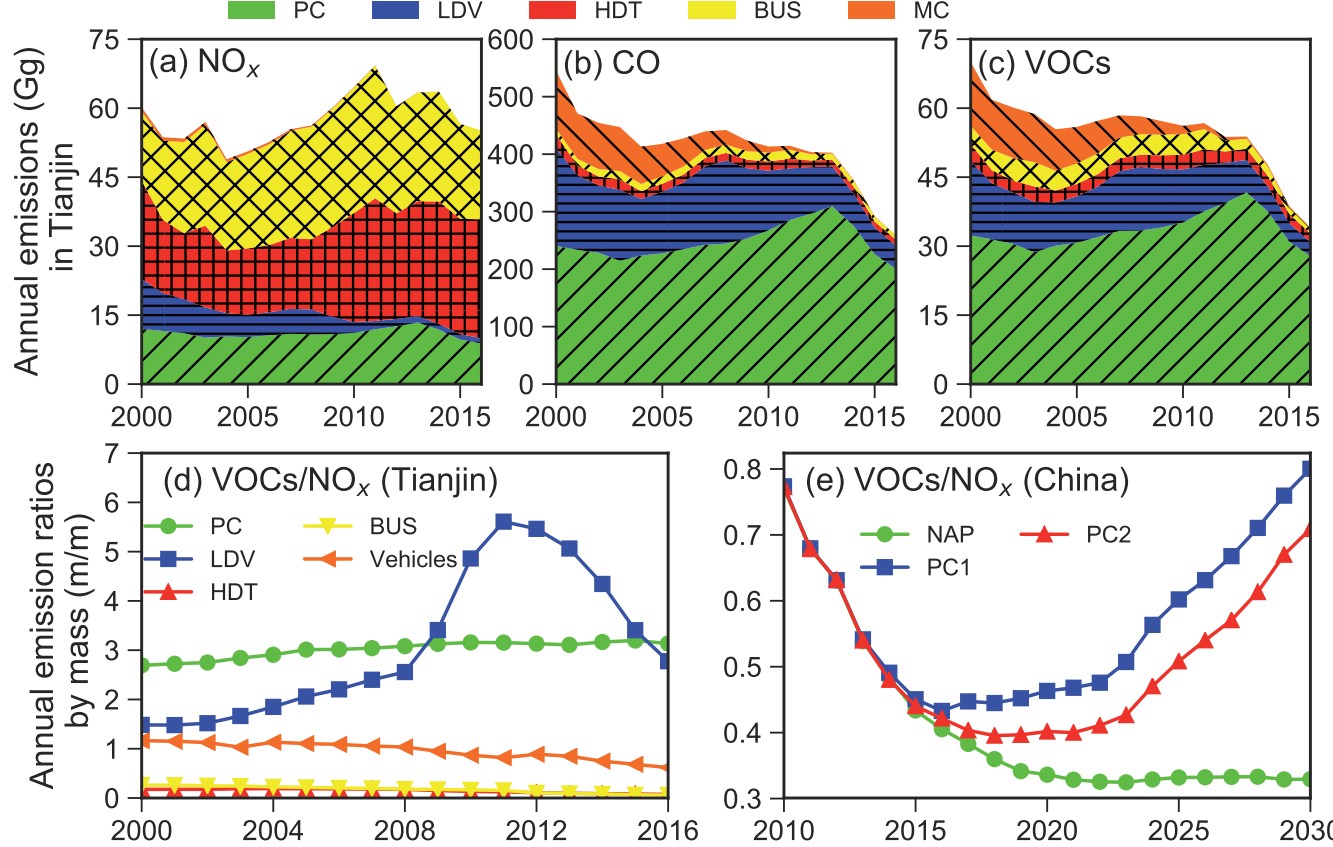

**Figure 11.** Estimated annual vehicle emissions of (a) $NO_X$, (b) CO, and (c) VOCs in Tianjin during 2000-2016. The vehicular emission ratios (m/m) of VOCs to $NO_x$ in (d) Tianjin and (e) China. (PC: passenger cars; LDV: light-duty vehicles; HDT: heavy-duty trucks; BUS: buses; MC: motocycles. NAP: Standards are assumed at the same level of 2013 during the future years (e.g., China IV for light-duty gasoline vehicles and China III for heavy-duty diesel vehicls); PC1: Increasingly stringent emission standards and improved fuel quality are implemented according to a probably timetable; PC2: For internal combustion engine vehicles, the implementation of future emission standards is consistent with that under the PC1 scenario. The vehicle emissions of (a) $NO_X$, (b) CO, and (c) VOCs in Tianjin during 2000-2016 were estimated by the COPERT IV model. The estimated total vehicular emission ratios of VOCs to $NO_x$ in China for NAP, PC1, and PC2 scenarios from 2010 to 2030 were based on Wu et al. (2017)'s study.)



**Table 1.** Average emission factors of Wujinglu tunnel in this study and their comparisons with other tunnel studies.

| City | Tunnel | Gradient | Test year | Vehicle type | Emission factors (mg veh$^{-1}$ km$^{-1}$) | | | |
|------|--------|----------|-----------|--------------|------|------|------|------|
| | | | | | NO | NO$_2$ | NO$_x$ | CO |
| Hong Kong[a] | ShingMun tunnel | 1% | 2004 | 50% DVs | N.A. | N.A. | 878±308 | 1845±434 |
| Taiwan[b] | Hsuehshan tunnel | -1.3% | 2006 | LDV | N.A. | N.A. | 145±67 | 910±470 |
| | | 1.3% | 2006 | LDV | N.A. | N.A. | 331±166 | 1470±630 |
| Shanghai[c] | East Yan'an Road tunnel | -3% | 2012 | 94.5% GVs | N.A. | N.A. | N.A. | 1266±889 |
| | | 3% | 2012 | 94.5% GVs | N.A. | N.A. | N.A. | 3353±2155 |
| Changsha[d] | Yingpan Road tunnel | -6% | 2013 | 96.3%-98.4% GVs | N.A. | N.A. | 121±22 | 754±561 |
| | | 6% | 2013 | 96.3%-98.4% GVs | N.A. | N.A. | 818±755 | 6050±5940 |
| Guangzhou[e] | Zhujiang tunnel | N.A. | 2013 | 88.2% GVs | N.A | N.A | 560±50 | N.A. |
| | | N.A. | 2014 | 61%LDV; 27% LPG; 12% HDV | N.A. | N.A. | 1286±204 | 3096±680 |
| Shanghai[f] | East Yan'an Road tunnel | N.A. | 2016 | 94.1% GVs | N.A | N.A | 400±250 | 1840±900 |
| Tianjin[g] | Wujinglu tunnel | -4% | 2017 | 94.2% GVs | 62±72 | 17±11 | 79±78 | 270±342 |
| | | 4% | 2017 | 94.2% GVs | 159±73 | 24±20 | 181±88 | 578±382 |
| | | Overall | 2017 | 94.2% GVs | 98±70 | 16±9 | 117±78 | 345±250 |

[a] Hong Kong (Cheng et al., 2006); [b] Taiwan (Chang et al., 2008); [c] Shanghai (Deng et al., 2015); [d] Changsha (Deng et al., 2015); [e] Guangzhou (Liu et al., 2014; Zhang et al., 2015); [f] Shanghai (Huang et al., 2017); [g] This study.

N.A. – not available.





**Table 2.** Typical ratios (v/v) related to vehicle emissions compared to previous tunnel studies.

| | Taiwan[a] | Hong Kong[b] | Guangzhou[c] | Taiwan[d] | Average[e] | Guangzhou[f] | Nanjing[g] | Hong Kong[h] | Tianjin[i] | Average[j] |
|---|---|---|---|---|---|---|---|---|---|---|
| | 2000 | 2003 | 2004 | 2005 | 2000-2005 | 2014 | 2015 | 2015 | 2017 | 2014-2017 |
| T/B | 2.01 | 2.26 | 1.44 | 4.19 | 2.48±1.19 | 2.03 | 2.16 | 3.51 | 1.42 | 2.28±0.88 |
| $i$-P/B | 1.11 | 1.35 | 2.40 | 7.39 | 3.06±2.94 | 4.00 | 5.24 | 1.93 | 3.74 | 3.73±1.36 |
| $i$-P/T | 0.55 | 0.60 | 1.67 | 1.76 | 1.15±0.66 | 1.97 | 2.43 | 0.55 | 2.64 | 1.9±0.94 |
| $i$-B/n-B | 0.70 | 0.63 | 0.50 | NA | 0.61±0.1 | 0.58 | 0.94 | 0.72 | 0.28 | 0.63±0.28 |
| $i$-P/n-P | 1.31 | 3.29 | 2.94 | 2.08 | 2.41±0.89 | 2.43 | 3.64 | 1.92 | 3.18 | 2.79±0.77 |
| MTBE/B | N.A. | N.A. | N.A. | N.A. | N.A. | N.A. | 0.38 | N.A. | 1.12 | 0.75±0.52 |
| MTBE/T | N.A. | N.A. | N.A. | N.A. | N.A. | N.A. | 0.18 | N.A. | 0.79 | 0.48±0.43 |
| B/2,2-DimB | 10.21 | 24.82 | 18.76 | 2.87 | 14.16±9.63 | 6.34 | 11.28 | 15.45 | 12.79 | 11.46±3.82 |
| T/2,2-DimB | 20.56 | 56.12 | 26.95 | 12.00 | 28.91±19.15 | 12.86 | 24.32 | 54.25 | 18.10 | 27.38±18.51 |
| E/2,2-DimB | 3.62 | 5.28 | 6.35 | 1.90 | 4.29±1.95 | 2.94 | 2.98 | 8.93 | 3.15 | 4.5±2.95 |
| $m,p$-X/2,3-DimB | 5.40 | 5.14 | 5.13 | 0.54 | 4.05±2.35 | 4.87 | 0.64 | 4.60 | 2.96 | 3.27±1.94 |
| $o$-X/2,2-DimB | 4.85 | 6.49 | 6.57 | 2.28 | 5.05±2.01 | 3.86 | 3.81 | 2.44 | 2.29 | 3.1±0.85 |
| $m,p$-X/E | 1.51 | 1.46 | 2.94 | 1.59 | 1.88±0.71 | 3.10 | 3.39 | 1.55 | 2.15 | 2.55±0.85 |
| B/E | 2.82 | 4.71 | 2.96 | 1.51 | 3±1.31 | 2.16 | 3.79 | 1.73 | 4.06 | 2.93±1.16 |
| T/E | 5.69 | 10.64 | 4.25 | 6.30 | 6.72±2.75 | 4.37 | 8.17 | 6.08 | 5.74 | 6.09±1.57 |

T: Toluene; B: Benzene; E: Ethylbenzene; $m,p$-X: $m,p$-Xylenes; $o$-X: $o$-xylene; $i$-P: $i$-Pentane; n-P: n-Pentane; $i$-B: $i$-Butane; n-B: n-Butane; MTBE: methyl tert-butyl ether; 2,2-DimB: 2,2-Dimethylbutane; 2,3-DimB: 2,3-Dimethylbutane; NA: not available.

[a] Taiwan (Hwa et al., 2002); [b] Hong Kong (Ho et al., 2009); [c] Guangzhou (Zhang et al., 2018c); [d] Taiwan (Hung-Lung et al., 2007); [e] Averaged values from tunnel studies in Taiwan in 2000, Hong Kong in 2003, Guangzhou in 2004, and Taiwan in 2005. [f] Guangzhou (Zhang et al., 2018c); [g] Nanjing (Zhang et al., 2018b); [h] Hong Kong in 2015 (Cui et al., 2018); [i] This study; [j] Averaged values from tunnel studies in Guangzhou in 2014, Nanjing in 2015, Hong Kong in 2015, and Tianjin in 2017. N.A. – not available.