# Peer review of "Vehicular volatile organic compounds (VOCs)- $NO_x$ -CO emissions in a tunnel study in northern China: emission factors, profiles, and source apportionment"

_Atmospheric Chemistry and Physics, 2018_

## Referee Comment (RC1) · Anonymous Referee #1 · 10 Jul 2018

The manuscript by Song C. et al is a study for primary emission factors (EFs) of vehicular volatile organic compounds (VOCs)-NOx-CO, and source profiles of vehicular VOCs through a well-designed tunnel test in a megacity of northern China. The fleet-average EFs and source profiles are limited in northern China because of few tunnel studies, which were difficult to organize. In this study, the EFs of VOCs from tailpipe and evaporative emissions resolved by PMF were obtained, which should be the key research highlight. The methods used in this study and corresponding conclusions are much helpful for tunnel tests and also for following researches on source apportionment, evaluation of EFs, emission inventory and control policy assessment. The paper is well written and structured and is full of high quality data. I suggest that the manuscript can be published after minor revision. Specific comments: (1) In the present manuscript, the paper presented ozone formation potential from fleet-average vehicle emissions. It could be better to further quantify the ozone or SOA formation potential from tailpipe and evaporative VOCs. The break-down knowledge for different sources (tailpipe and evaporative VOC emissions) will provide new insights into useful emission control strategies. (2) The resolved source profiles of tailpipe and evaporative VOCs by PMF should be compared with other published studies. (3) Accurate measurement of the fluxes of inlet and outlet is essential to the calculation of EFs. It would be useful to have a brief description of the methods used to measure the fluxes. (4) Is it possible to resolve tailpipe and evaporative VOCs by the method of chemical mass balance (CMB)? (5) I suggest to conduct the similar study in other seasons with different ranges of ambient temperature.

---

## Referee Comment (RC2) · Anonymous Referee #2 · 11 Jul 2018

The tunnel based emission factors in Tianjin has been reported. The new information provided in this manuscript is not enough for a separate paper. Please reference to: Heavy-duty diesel vehicles dominate vehicle emissions in a tunnel study in northern China, Science of The Total Environment, Volumes 637–638, 2018, Pages 431-442.

———————————————

---

## Author Comment (AC1) · 18 Aug 2018

**Authors' Reponses to Reviewer Comments**

Congbo Song*

E-mail: songcongbo@163.com

MS No.: acp-2018-387

Title: Vehicular volatile organic compounds (VOCs)-NO$_x$-CO emissions in a tunnel study in northern China: emission factors, profiles, and source apportionment

**Referee #1**

**General comments:**

The manuscript by Song C. et al is a study for primary emission factors (EFs) of vehicular volatile organic compounds (VOCs)-NOx-CO, and source profiles of vehicular VOCs through a well-designed tunnel test in the northern China. The fleet-average EFs and source profiles are limited in the northern China because of few tunnel studies. In this study, the EFs of VOCs from tailpipe and evaporative emissions resolved by PMF were obtained, which should be the key research highlight. The methods used in this study and corresponding conclusions are much helpful for tunnel tests and also for following source apportionment, evaluation of EFs, emission inventory and control policy assessment researches. The paper is well written and structured and is full of high quality data. I suggest that the manuscript needs minor revision before published.

**General response:**

Thanks a lot for your positive comments on this manuscript. We have improved this manuscript by answering the reviewers' comments and suggestions.

**Specific comments:**

**Q1:**

In the present manuscript, the paper presented ozone formation potential (OFP) from fleet-average vehicle emissions. It could be better to further quantify the ozone or SOA formation potential from tailpipe and evaporative VOCs. The break-down knowledge for different sources (tailpipe and evaporative VOC emissions) will provide insights into useful emission control strategies.

**A1:**

In our previous manuscript, we estimated the average emission factors for 30 species from tailpipe and evaporative VOCs in Table S5. In the revised manuscript ,we made some changes to reduce the 30 species to 20 species. To minimize the uncertainties caused by the NMVOC species with low $\Delta$NMVOCs values, the top 30 NMVOC species from vehicular emissions were considered in source apportionment by PMF. Additionally, the species with a number of samples of non-negative $\Delta$NMVOCs (i.e., $C_{Outlet} - C_{Inlet}$>0) less than 75% were excluded from the PMF model. Finally, the 20 NMVOC species (Table 1, see below) were introduced into the PMF model.

As listed in Table 1, the EFs of the top 20 VOC compounds in vehicular evaporative and tailpipe emissions were estimated via relative contributions (Fig.1a) and fleet-average EFs. The EFs for vehicular evaporative and tailpipe emissions were estimated to be 24.9±7.5 and 15.7±3.3 mg km$^{-1}$ veh$^{-1}$, respectively. In addition, the EFs and OFP from vehicular evaporative and tailpipe emissions were 60.3±4.1 and 74.9±4.5 mg km$^{-1}$ veh$^{-1}$, respectively. During the measurement campaign, evaporative emissions accounted for over half (62.7±37.3%) of the total vehicular NMVOC emissions, and nearly half (44.6%) of the total vehicular OFP.

Thank you for your kind advice. The relevent contens have been added to the revised manuscript.

Table 1: The slope and $R^2$ from the linear regression between the measured and modeled VOCs concentrations for 2-factor solution, and the estimated EFs (mg km$^{-1}$ veh$^{-1}$) of VOCs from evaportative and tailpipe emissions.

| Compound | Slope | $R^2$ | EFs (mg km$^{-1}$ veh$^{-1}$) | |
| --- | --- | --- | --- | --- |
| | | | Evaporative | Tailpipe |
| Ethylene | 1.02 | 0.83 | 0.99±0.18 | 1.57±0.28 |
| Isopentane | 0.98 | 0.83 | 6.19±2.14 | 0.62±0.2 |
| Ethane | 0.84 | 0.76 | 1.85±0.82 | 0.07±0.03 |
| Toluene | 0.91 | 0.72 | 0.88±0.3 | 2.42±0.79 |
| n-Butane | 0.99 | 0.87 | 1.71±0.75 | 0.3±0.16 |
| n-Pentane | 0.99 | 0.95 | 1.93±0.9 | 0.22±0.1 |
| MTBE | 0.79 | 0.86 | 1.43±0.7 | 1.06±0.5 |
| Acetylene | 0.76 | 0.45 | 0.1±0.03 | 0.59±0.16 |
| Propylene | 0.92 | 0.70 | 0.36±0.08 | 0.67±0.14 |
| Benzene | 0.99 | 0.87 | 1.16±0.31 | 0.78±0.2 |
| 2-Methylpentane | 0.97 | 0.95 | 1.1±0.41 | 0.5±0.18 |
| 3-Methylpentane | 0.90 | 0.94 | 0.93±0.37 | 0.46±0.17 |
| m/p-Xylene | 0.79 | 0.67 | 0.04±0.01 | 1.39±0.4 |
| Propanal | 0.50 | 0.44 | 0.52±0.29 | 0.12±0.06 |
| n-Hexane | 0.82 | 0.93 | 0.48±0.15 | 0.38±0.12 |
| Isobutane | 0.89 | 0.65 | 0.51±0.25 | 0.05±0.02 |
| 2,2,4-Trimethylpentane | 1.00 | 0.92 | 0.07±0.01 | 0.6±0.09 |
| Ethylbenzene | 0.70 | 0.58 | 0.08±0.03 | 0.58±0.22 |
| 3-Methylhexane | 0.92 | 0.97 | 0.26±0.07 | 0.29±0.08 |
| 2,3-Dimethylbutane | 0.91 | 0.94 | 0.29±0.11 | 0.11±0.04 |
| NMVOCs | 0.98 | 0.98 | 24.89±7.52 | 15.67±3.32 |

[Figure]

Figure 1: The source profiles (% of species, represented by scatters) and contributions (Conc. of Species, represented by bars) of 2-factor (a: Factor1, b: Factor2) solution by the PMF model. Factor1 and Factor2 was identified as evaporative and tailpipe emissions, respectively.

**Q2:**

The resolved source profiles of tailpipe and evaporative VOCs by PMF should be compared with other published studies.

**A2:**

Thank you for your kind suggestion. In the present study, we aimed at providing new methods of source apportionment of on-road vehicular VOC emissions. It is of great important to quantify the relative contribution of tailpipe and evaporative emissions to on-road vehicular VOC emissions. There were some uncertainties underlying the resolved source profiles by PMF model as there were limited samples. In addition, We discussed the source identification of the resolved factors in details in the manuscript from the perspective of both the tracers and the temperature dependence. We belive that the resolved factors (Figure 1) could be identified as tailpipe and evaporative emissions.

**Q3:**

Accurate measurement of the fluxes of inlet and outlet is essential to the calculation of EFs. It would be useful to have a brief description of the methods used to measure the fluxes.

**A3:**

Thank you for your suggestion. In this study, the actual volumetric flow rates induced by the vehicular fleet and the prevailing winds in the tunnel were continuously measured using ultrasonic gas flowmeters (Flowsick-200 SICK MAIHAK, Germany) with a time resolution of 1 min, which have also been used in previous tunnel tests[1–3].

The FLOWSIC 200 operates by measuring the transit delay of an ultrasonic pulse. Sender/receive unites are mounted on both sides of the tunnel at a certain angle to the flow direction (Figure 2).

The measuring path L is equal to the active measuring distance, that is, the area through which the air flows. Given the measuring path L, sound velocity c, and the angle of inclination a between the sound and flow direction, the sound transit time when the signal is transmitted in the direction of the flow (forward direction) can be expressed as:

$$t_v = \frac{L}{c + v \cdot cos\alpha} \tag{1}$$

The sound transmit against the direction of flow:

$$t_r = \frac{L}{c - v \cdot cos\alpha} \tag{2}$$

Conversation to $v$:

$$v = \frac{L}{2 \cdot cos\alpha} \cdot (\frac{1}{t_v} - \frac{1}{t_r}) \tag{3}$$

[Figure]

Figure 2: FLOWSIC 200 operating principle.

**Q4:**

Is it possible to resolve tailpipe and evaporative VOCs by the method of chemical mass balance (CMB)?

**A4:**

Thank you for your suggestion. Vehicular emissions in the tunnel consisted of tailpipe and evaporative emissions. Profiles of tailpipe and evaporative emissions are necessary for CMB analysis to apportion these two factors. Vehicular evaporative emissions can be generally grouped into hot soak, diurnal, permeation, refueling process, and running loss[4,5]. However, source profiles and EFs of VOC from hot soak, diurnal, permeation and refueling process have been well characterized in laboratories, but VOC emissions from running loss are less understood because no facility in China could accommodate the test procedures or track-based tank temperature profile generation[5]. Thus, the source profile of evaporative emissions could be hardly obtained since the limited information for running loss. Thus, the CMB method is not ideally suitable for source aportionment of on-road vehicular VOC emissions in this case.

Only if we could quantify running losses in the laboratory test, we could conduct CMB analysis for source aportionment of on-road vehicular VOC emissions. Moreover, our study provide a detailed experiments and mathods to quantify the relation contribution of tailpipe and evaporative emissions to on-road vehicular VOC emissions.

**Q5:**

I suggest to conduct the similar study in other seasons with different ranges of ambient temperature.

**A5:**

Thank you for your suggestion. Vehicular emissions are greatly influence by ambient temperature, mainly consisting of cold-start and evaporative emissions. In this study, we quantified the temperature-dependent evaporative emissions (Figure 3). It is very meaningful to conduct the similar study in other seasons with different ranges of ambient temperature.

However, it is very hard to conduct such campagin considering the safety and permission. We will try to work on it in the future.

[Figure]

Figure 3: The linear regressions (filled with 95% regression confidence intervals) between emission factors of (a) Isopentane, (b) n-Butane, (c) n-Pentane, and (d) Isobutane against ambient temperature data.

**References**

(1) Song, C.; Ma, C.; Zhang, Y.; Wang, T.; Wu, L.; Wang, P.; Liu, Y.; Li, Q.; Zhang, J.; Dai, Q.; Zou, C.; Sun, L.; Mao, H. Heavy-duty diesel vehicles dominate vehicle emissions

in a tunnel study in northern China. *Science of The Total Environment* **2018**, *637-638*, 431 – 442.

(2) Zhang, Q.; Wu, L.; Fang, X.; Liu, M.; Zhang, J.; Shao, M.; Lu, S.; Mao, H. Emission factors of volatile organic compounds (VOCs) based on the detailed vehicle classification in a tunnel study. *Science of The Total Environment* **2018**, *624*, 878 – 886.

(3) Imhof, D.; Weingartner, E.; Prévôt, A. S. H.; Ordóñez, C.; Kurtenbach, R.; Wiesen, P.; Rodler, J.; Sturm, P.; McCrae, I.; Ekström, M.; Baltensperger, U. Aerosol and $NO_x$ emission factors and submicron particle number size distributions in two road tunnels with different traffic regimes. *Atmospheric Chemistry and Physics* **2006**, *6*, 2215–2230.

(4) Yue, T.; Yue, X.; Chai, F.; Hu, J.; Lai, Y.; He, L.; Zhu, R. Characteristics of volatile organic compounds (VOCs) from the evaporative emissions of modern passenger cars. *Atmospheric Environment* **2017**, *151*, 62 – 69.

(5) Liu, H.; Man, H.; Tschantz, M.; Wu, Y.; He, K.; Hao, J. VOC from Vehicular Evaporation Emissions: Status and Control Strategy. *Environmental Science & Technology* **2015**, *49*, 14424–14431.

**Vehicular volatile organic compounds (VOCs)-NO$_x$-CO emissions in a tunnel study in northern China: emission factors, profiles, and source apportionment**

Congbo Song[1], Yan Liu[1], Shida Sun[1], Luna Sun[1], Yanjie Zhang[1], Chao Ma[1], Jianfei Peng[2], Qian Li[1], Jinsheng Zhang[1], Qili Dai[1], Baoshuang Liu[1], Peng Wang[2], Yi Zhang[1], Ting Wang[1], Lin Wu[1], Min Hu[3], and Hongjun Mao[1]

[1]Center for Urban Transport Emission Research & State Environmental Protection Key Laboratory of Urban Ambient Air Particulate Matter Pollution Prevention and Control, College of Environmental Science and Engineering, Nankai University, Tianjin, 300071, China
[2]Zachry Department of Civil Engineering, Texas A and M University, College Station, TX, 77845, USA
[3]Laboratory of Environmental Simulation and Pollution Control, College of Environmental Sciences and Engineering, Peking University, Beijing, 100871, China

**Correspondence:** Hongjun Mao (hongjunm@nankai.edu.cn)

**Abstract.** Vehicular emission is a key contributor to ambient volatile organic compounds (VOCs) and NO$_x$ in Chinese megacities. However, the information of real-world emission factors (EFs) for a typical urban fleet is still limited, hindering the development of a more reliable emission inventory in China. Based on a more-than-two-week (August 8-24, 2017) tunnel test in urban Tianjin in northern China, and on the use of a statistical regression model, the Positive Matrix Factorization (PMF) receptor model, and the Calculate Emissions from Road Transport (COPERT) IV model, characteristics of vehicular VOCs-NO$_x$-CO emissions were analyzed systematically. The fleet-average EFs (pollutant: downslope, upslope, and overall in mg km$^{-1}$ veh$^{-1}$) were estimated respectively as follows: (NO: 61.92±72.46, 158.58±73.48, 97.52±69.84), (NO$_2$: 16.52±11.49, 23.98±20.14, 15.86±9.38), (NO$_x$: 79.45±78.43, 181.22±88.29, 116.56±77.61), and (CO: 269.96±342.38, 577.76±382.22, 344.67±250.01). The EFs of NO-NO$_2$-NO$_x$ and CO from heavy-duty vehicles (or diesel vehicles) were differentiated from light-duty vehicles (or gasoline vehicles). The ratios (v/v) of NO$_2$ to NO$_x$ in the primary vehicular exhaust were approximately 0.18±0.09, 0.10±0.22 and 0.10±0.05 for downslope, upslope, and the entire tunnel, respectively. The fleet-average EF of the 99-target non-methane VOCs (NMVOCs) was 40.56±12.18 mg km$^{-1}$ veh$^{-1}$, lower than the previous studies in China. The BTEX (benzene, toluene, ethylbenzene, $p$-xylene, $m$-xylene and $o$-xylene) levels decreased by approximately 79% when emission standards increased from China I to China V. The  vehicular T/B emission ratio (ER, v/v) was 1.42±0.33. Isopentane could be employed as a suitable indicator of vehicular emission. The characteristic ratios (v/v) of C6-C8 aromatics to isopentane from vehicular emissions were 0.29±0.07 (benzene), 0.42±0.19 (toluene), 0.17±0.08 ($m,p$-xylene), 0.06±0.03 ($o$-xylene), 0.01±0.00 (styrene), and 0.08±0.06 (ethylbenzene), respectively. Evaporative (mostly running loss) and tailpipe VOC emissions in tunnel tests could be resolved by the PMF model. The evaporative emissions accounted for nearly one-half of the total vehicular VOC emissions and ozone formation poential, indicating that evaporative and tailpipe emissions contributed equally to NMVOC

emission inventories, air quality, and energy. The relative contributions of evaporative NMVOC emissions to total vehicular NMVOC emissions are temperature-dependent with the average increasing ratio of 4% °C$^{-1}$. The primary  ER (m/m) of VOCs/NO$_x$ was approximately 2.04, suggesting that vehicular NO$_x$ and VOCs can be co-emitted with a proper ER. According to the vehicular ERs of VOCs/NO$_x$ in Tianjin (2000-2016) and China (2010-2030), as even more stringent emission standards are implemented in the future, the O$_3$ chemical regimes were likely to be VOCs-limited (i.e., 8:1 threshold) for cities or regions where VOCs and NO$_x$ emissions are dominated by vehicular exhaust. Our study enriched the database on the fleet-average emission factors of on-road vehicles for emission inventory, air quality modeling, and health effects studies, provided diagnositc ratios (v/v) of primarily vehicular emissions for apportioning vehicular and non-vehicular contributions to reactive species and 
[revised manuscript text omitted]

30   in source apportionment by PMF. Additionally, the species with a number of samples of non-negative ΔNMVOCs (i.e., $C_{Outlet} - C_{Inlet}$>0) less than 75% were excluded from the PMF model. Finally, the 20 NMVOC species (Fig. 7) were introduced into the PMF model. The highly reactive NMVOCs species were not excluded in the source apportionment because of the low chemical reactions in the tunnel where there is weaker solar radiation, which is different from traditional source apportionment of ambient VOCs (Zheng et al., 2018; Liu et al., 2016).

Two factors (Factor1 and Factor2) were resolved by the PMF analysis. Approximately 98% of the measured NMVOCs from vehicular emissions were explained using the PMF (Fig.A4a). Moreover, for the top  20 individual NMVOCs species, the PMF model also reproduced the predicted values well, with the  $R^2$  ranging from 0.44 to 0.97 (Table.A5).

5 Therefore, we considered that the ΔNMVOCs concentrations in the tunnel could be resolved by the two factors using the PMF model.

The fleet-average EFs of isopentane, n-butane, n-pentane, and  isobutane generally showed good correlations with ambient temperature data (Fig.8), suggesting that these four compounds could be recognized as primary indicators of evaporation emissions (Yue et al., 2017; Liu et al., 2008; Tsai et al., 2006; Hwa et al., 2002). High loadings of isopentane

10 (90.9%), n-butane (84.1%), n-pentane (89.9%), and isobutane (91.5%) were found in Factor1. The source apportionment of the NMVOC samples at 3-h intervals provide a unique opportunity to discuss the diurnal variations in factor contributions (Fig.A4b). Factor1 had a higher contribution to vehicular VOC emissions during the daytime, and a lower contribution during the nighttime. In addition, diurnal variations in Factor1 contribution (%) linearly correlated with diurnal variations in temperatures ($y = 0.04x(°C) - 0.61, R^2 = 0.722$).

15 Therefore, Factor1 was identified as evaporative emissions, and Factor2 was identified as tailpipe emissions.

As listed in Table.A5, the EFs of the top  20 VOC compounds in vehicular evaporative and tailpipe emissions were estimated via relative contributions (Fig.7a) and fleet-average EFs (Fig.A4). The EFs for vehicular evaporative and tailpipe emissions were estimated to be 24.9±7.5 and 15.7±3.3 mg km$^{-1}$ veh$^{-1}$, respectively. In addition, the EFs of OFP from vehicular evaporative and tailpipe emissions were 60.3±4.1 and 74.9±4.5 mg km$^{-1}$ veh$^{-1}$, respectively.

20 During the measurement campaign, evaporative emissions accounted for over half (62.7± 37.3%) of the total vehicular NMVOC emissions, and nearly half (44.6%) of the total vehicular OFP. However, the tailpipe emissions of NMVOCs might be underestimated due to the downslope (1#-2#) road conditions. A previous tunnel study noted that the upslope EF of NMVOCs is only 1.3 times that of the downslope (Chang et al., 2008). Thus, the upslope evaporative emissions were estimated to account for over one-half (56.4%) of the total vehicular upslope NMVOC emissions,

[revised manuscript text omitted]
 vehicular T/B emission ratio (ER) was 1.42±0.33. Isopentane could be employed as a suitable indicator of vehicular emission. The characteristic ratios of C6-C8 aromatics to isopentane from vehicular emissions were 0.29±0.07 (benzene), 0.42±0.19 (toluene), 0.17±0.08 ($m,p$-xylene), 0.06±0.03 ($o$-xylene), 0.01±0.00 (styrene), and 0.08±0.06 (ethylbenzene), respectively. Evaporative (mostly running loss) and tailpipe VOC emissions in tunnel tests could be resolved by the PMF

model, and  evaporative emissions accounted for over half (62.7±37.3%) of the total vehicular NMVOC emissions, and nearly half (44.6%) of the total vehicular OFP. Besides, the relative contributions of evaporative NMVOC emissions to total vehicular NMVOC emissions are found to be temperature-dependent with increasing ratio of 4% $°C^{-1}$.

[revised manuscript text omitted]

**Figure 9.** The BTEX (Benzene, Ethylbenzene, Toluene, *o*-Xylene, and *m/p*-Xylene) levels (mean±standard deviation) observed in different tunnel studies (Taiwan in 2000 (Hwa et al., 2002); Hong Kong in 2003 (Ho et al., 2009); Guangzhou in 2004 (Zhang et al., 2018c); Taiwan in 2005 (Hung-Lung et al., 2007); Guangzhou in 2014 (Zhang et al., 2018c); Nanjing in 2015 (Zhang et al., 2018b); Hong Kong in 2015 (Cui et al., 2018)), and the linear regressions (dash-dot lines filled with 95% regression confidence intervals) between the emission factors of BTEX and the test years.

[Figure]

**Figure 10.** (a)The diurnal variations (filled with one standard deviation) of the NMVOCs/NO$_x$ in the inlet and outlet of the tunnel, and (b) the linear regression (filled with 95% confidence intervals) between ΔNMVOCs (i.e., $C_{Outlet} - C_{Inlet}$) and ΔNO$_x$ (i.e., $C_{Outlet} - C_{Inlet}$), ΔNMVOCs (ppbv)=0.18×ΔNO$_x$ (ppbv)+5.80, ΔNMVOCs (ppbC)=0.84×ΔNO$_x$ (ppbv)+22.16, ΔNMVOCs ($\mu$g m$^{-3}$)=2.04×ΔNO$_x$ ($\mu$g m$^{-3}$)+26.59.

[Figure]

**Figure 11.** Estimated annual vehicular emissions of (a) $NO_X$, (b) CO, and (c) VOCs in Tianjin during 2000-2016. The vehicular emission ratios (m/m) of VOCs to $NO_x$ in (d) Tianjin and (e) China. (PC: passenger cars; LDV: light-duty vehicles; HDT: heavy-duty trucks; BUS: buses; MC: motocycles. NAP: Standards are assumed at the same level of 2013 during the future years (e.g., China IV for light-duty gasoline vehicles and China III for heavy-duty diesel vehicls); PC1: Increasingly stringent emission standards and improved fuel quality are implemented according to a probably timetable; PC2: For internal combustion engine vehicles, the implementation of future emission standards is consistent with that under the PC1 scenario. The vehicle emissions of (a) $NO_x$, (b) CO, and (c) VOCs in Tianjin during 2000-2016 were estimated by the COPERT IV model. The estimated total vehicular emission ratios of VOCs to $NO_x$ in China for NAP, PC1, and PC2 scenarios from 2010 to 2030 were based on Wu et al. (2017)'s study.)

**Table 1.** Average emission factors of Wujinglu tunnel in this study and their comparisons with other tunnel studies.

| City | Tunnel | Gradient | Test year | Vehicle type | Emission factors (mg veh$^{-1}$ km$^{-1}$) | | | |
|------|--------|----------|-----------|--------------|------|------|------|------|
| | | | | | NO | NO$_2$ | NO$_x$ | CO |
| Hong Kong[a] | ShingMun tunnel | 1% | 2004 | 50% DVs | N.A. | N.A. | 878±308 | 1845±434 |
| Taiwan[b] | Hsuehshan tunnel | -1.3% | 2006 | LDV | N.A. | N.A. | 145±67 | 910±470 |
| | | 1.3% | 2006 | LDV | N.A. | N.A. | 331±166 | 1470±630 |
| Shanghai[c] | East Yan'an Road tunnel | -3% | 2012 | 94.5% GVs | N.A. | N.A. | N.A. | 1266±889 |
| | | 3% | 2012 | 94.5% GVs | N.A. | N.A. | N.A. | 3353±2155 |
| Changsha[d] | Yingpan Road tunnel | -6% | 2013 | 96.3%-98.4% GVs | N.A. | N.A. | 121±22 | 754±561 |
| | | 6% | 2013 | 96.3%-98.4% GVs | N.A. | N.A. | 818±755 | 6050±5940 |
| Guangzhou[e] | Zhujiang tunnel | N.A. | 2013 | 88.2% GVs | N.A | N.A | 560±50 | N.A. |
| | | N.A. | 2014 | 61%LDV; 27% LPG; 12% HDV | N.A. | N.A. | 1286±204 | 3096±680 |
| Shanghai[f] | East Yan'an Road tunnel | N.A. | 2016 | 94.1% GVs | N.A | N.A | 400±250 | 1840±900 |
| Tianjin[g] | Wujinglu tunnel | -4% | 2017 | 94.2% GVs | 62±72 | 17±11 | 79±78 | 270±342 |
| | | 4% | 2017 | 94.2% GVs | 159±73 | 24±20 | 181±88 | 578±382 |
| | | Overall | 2017 | 94.2% GVs | 98±70 | 16±9 | 117±78 | 345±250 |

[a] Hong Kong (Cheng et al., 2006); [b] Taiwan (Chang et al., 2008); [c] Shanghai (Deng et al., 2015); [d] Changsha (Deng et al., 2015); [e] Guangzhou (Liu et al., 2014; Zhang et al., 2015); [f] Shanghai (Huang et al., 2017); [g] This study.

N.A. – not available.

**Table 2.** Typical ratios (v/v) related to vehicle emissions compared to previous tunnel studies.

[revised manuscript text omitted]

---

## Author Comment (AC2) · 18 Aug 2018

**Authors' Reponses to Reviewer Comments**

Congbo Song*

E-mail: songcongbo@163.com

MS No.: acp-2018-387

Title: Vehicular volatile organic compounds (VOCs)-NO$_x$-CO emissions in a tunnel study in northern China: emission factors, profiles, and source apportionment

**Referee #2**

**General comments:**

The tunnel based emission factors in Tianjin has been reported. The new information provided in this manuscript is not enough for a separate paper. Please reference to: Heavy-duty diesel vehicles dominate vehicle emissions in a tunnel study in northern China, Science of The Total Environment, Volumes 637–638, 2018, Pages 431-442.

**General response:**

Thanks a lot for your reviewing. In our previous study (Heavy-duty diesel vehicles dominate vehicle emissions in a tunnel study in northern China, Science of The Total Environment, Volumes 637–638, 2018, Pages 431-442.), we estimated vehicular emissions (especially for heavy-duty diesel vehicles (HDDVs) and non-HDDVs) based on the emission factors derived from the same measurement campaign. However, the data, contents, and the results and discussion in this manuscript are totally different with our previous study.

(1) The emission factors for PM$_{2.5}$, NO, NO$_2$, NO$_x$, and CO are measured only under downslope road condition (1#-2#) in our previous study. In this manuscript, we compared the emission factors under upslope (2#-3#) road condition with those under downslope (1#-

2#) road condtion. In addition, we analyzed the impacts of road condition on the fractions of $NO_2$ in $NO_x$ ($f_{NO_2}$, v/v) from vehicular emissions.

(2) Vehicular VOC emissions (including source profiles and emission factors) from the measurement campaign were first characterized in this manuscript. Moreover, we reviewed available emission factors of VOCs in tunnel tests from 2000 (before China I) to 2017 (mainly China IV and V) to show the interannual changes of EFs of VOCs with the improvement of fuel quality.

(3) The highlights of this study could be the experiment design and methods of source apportionment of vehicular VOC emissions. With the control of tailpipe exhaust emissions, the contribution of evaporative emissions to vehicular VOC emissions is rapidly increasing. However, the relative importance of tailpipe and evaporative emissions were still less well kown in China. Our study suggested that the EFs and ozone formation potential (OFP) from vehicular evaporative and tailpipe emissions were 60.3±4.1 and 74.9±4.5 mg $km^{-1}$ $veh^{-1}$, respectively. During the measurement campaign, evaporative emissions accounted for over half (62.7±37.3%) of the total vehicular NMVOC emissions, and nearly half (44.6%) of the total vehicular OFP. The evaporative and tailpipe emissions contribute equally to NMVOC emission inventories, air quality, and energy. The results from this study could be very helpful for emission control of vehicular VOCs.

(4) This manuscript focused not only on the emission factors, but the partition of vehicular $NO_x$ emissions, typical diagnostic ratios, and vehicular emission ratio of VOCs to $NO_x$.

In summary, the information and stories included in our previous study and this manuscript are too abundant for a single paper. It could be appropriate to separate them. In addition, We have made some changes and improved this manuscript according to the reviewers' comments and suggestions.

**Vehicular volatile organic compounds (VOCs)-NO$_x$-CO emissions in a tunnel study in northern China: emission factors, profiles, and source apportionment**

Congbo Song[1], Yan Liu[1], Shida Sun[1], Luna Sun[1], Yanjie Zhang[1], Chao Ma[1], Jianfei Peng[2], Qian Li[1], Jinsheng Zhang[1], Qili Dai[1], Baoshuang Liu[1], Peng Wang[2], Yi Zhang[1], Ting Wang[1], Lin Wu[1], Min Hu[3], and Hongjun Mao[1]

[1]Center for Urban Transport Emission Research & State Environmental Protection Key Laboratory of Urban Ambient Air Particulate Matter Pollution Prevention and Control, College of Environmental Science and Engineering, Nankai University, Tianjin, 300071, China

[2]Zachry Department of Civil Engineering, Texas A and M University, College Station, TX, 77845, USA

[3]Laboratory of Environmental Simulation and Pollution Control, College of Environmental Sciences and Engineering, Peking University, Beijing, 100871, China

**Correspondence:** Hongjun Mao (hongjunm@nankai.edu.cn)

**Abstract.** Vehicular emission is a key contributor to ambient volatile organic compounds (VOCs) and NO$_x$ in Chinese megacities. However, the information of real-world emission factors (EFs) for a typical urban fleet is still limited, hindering the development of a more reliable emission inventory in China. Based on a more-than-two-week (August 8-24, 2017) tunnel test in urban Tianjin in northern China, and on the use of a statistical regression model, the Positive Matrix Factorization (PMF) receptor model, and the Calculate Emissions from Road Transport (COPERT) IV model, characteristics of vehicular VOCs-NO$_x$-CO emissions were analyzed systematically. The fleet-average EFs (pollutant: downslope, upslope, and overall in mg km$^{-1}$ veh$^{-1}$) were estimated respectively as follows: (NO: 61.92±72.46, 158.58±73.48, 97.52±69.84), (NO$_2$: 16.52±11.49, 23.98±20.14, 15.86±9.38), (NO$_x$: 79.45±78.43, 181.22±88.29, 116.56±77.61), and (CO: 269.96±342.38, 577.76±382.22, 344.67±250.01). The EFs of NO-NO$_2$-NO$_x$ and CO from heavy-duty vehicles (or diesel vehicles) were differentiated from light-duty vehicles (or gasoline vehicles). The ratios (v/v) of NO$_2$ to NO$_x$ in the primary vehicular exhaust were approximately 0.18±0.09, 0.10±0.22 and 0.10±0.05 for downslope, upslope, and the entire tunnel, respectively. The fleet-average EF of the 99-target non-methane VOCs (NMVOCs) was 40.56±12.18 mg km$^{-1}$ veh$^{-1}$, lower than the previous studies in China. The BTEX (benzene, toluene, ethylbenzene, $p$-xylene, $m$-xylene and $o$-xylene) levels decreased by approximately 79% when emission standards increased from China I to China V. The  vehicular T/B emission ratio (ER, v/v) was 1.42±0.33. Isopentane could be employed as a suitable indicator of vehicular emission. The characteristic ratios (v/v) of C6-C8 aromatics to isopentane from vehicular emissions were 0.29±0.07 (benzene), 0.42±0.19 (toluene), 0.17±0.08 ($m,p$-xylene), 0.06±0.03 ($o$-xylene), 0.01±0.00 (styrene), and 0.08±0.06 (ethylbenzene), respectively. Evaporative (mostly running loss) and tailpipe VOC emissions in tunnel tests could be resolved by the PMF model. The evaporative emissions accounted for nearly one-half of the total vehicular VOC emissions and ozone formation poential, indicating that evaporative and tailpipe emissions contributed equally to NMVOC

emission inventories, air quality, and energy. The relative contributions of evaporative NMVOC emissions to total vehicular NMVOC emissions are temperature-dependent with the average increasing ratio of 4% °C$^{-1}$. The primary  ER (m/m) of VOCs/NO$_x$ was approximately 2.04, suggesting that vehicular NO$_x$ and VOCs can be co-emitted with a proper ER. According to the vehicular ERs of VOCs/NO$_x$ in Tianjin (2000-2016) and China (2010-2030), as even more stringent emission standards are implemented in the future, the O$_3$ chemical regimes were likely to be VOCs-limited (i.e., 8:1 threshold) for cities or regions where VOCs and NO$_x$ emissions are dominated by vehicular exhaust. Our study enriched the database on the fleet-average emission factors of on-road vehicles for emission inventory, air quality modeling, and health effects studies, provided diagnositc ratios (v/v) of primarily vehicular emissions for apportioning vehicular and non-vehicular contributions to reactive species and 
[revised manuscript text omitted]
 vehicular T/B emission ratio (ER) was 1.42±0.33. Isopentane could be employed as a suitable indicator of vehicular emission. The characteristic ratios of C6-C8 aromatics to isopentane from vehicular emissions were 0.29±0.07 (benzene), 0.42±0.19 (toluene), 0.17±0.08 ($m,p$-xylene), 0.06±0.03 ($o$-xylene), 0.01±0.00 (styrene), and 0.08±0.06 (ethylbenzene), respectively. Evaporative (mostly running loss) and tailpipe VOC emissions in tunnel tests could be resolved by the PMF

model, and   evaporative emissions accounted for over half (62.7±37.3%) of the total vehicular NMVOC emissions, and nearly half (44.6%) of the total vehicular OFP. Besides, the relative contributions of evaporative NMVOC emissions to total vehicular NMVOC emissions are found to be temperature-dependent with increasing ratio of 4% °C$^{-1}$.

[revised manuscript text omitted]

**Figure 9.** The BTEX (Benzene, Ethylbenzene, Toluene, *o*-Xylene, and *m/p*-Xylene) levels (mean±standard deviation) observed in different tunnel studies (Taiwan in 2000 (Hwa et al., 2002); Hong Kong in 2003 (Ho et al., 2009); Guangzhou in 2004 (Zhang et al., 2018c); Taiwan in 2005 (Hung-Lung et al., 2007); Guangzhou in 2014 (Zhang et al., 2018c); Nanjing in 2015 (Zhang et al., 2018b); Hong Kong in 2015 (Cui et al., 2018)), and the linear regressions (dash-dot lines filled with 95% regression confidence intervals) between the emission factors of BTEX and the test years.

[Figure]

**Figure 10.** (a)The diurnal variations (filled with one standard deviation) of the NMVOCs/NO$_x$ in the inlet and outlet of the tunnel, and (b) the linear regression (filled with 95% confidence intervals) between ΔNMVOCs (i.e., $C_{Outlet} - C_{Inlet}$) and ΔNO$_x$ (i.e., $C_{Outlet} - C_{Inlet}$), ΔNMVOCs (ppbv)=0.18×ΔNO$_x$ (ppbv)+5.80, ΔNMVOCs (ppbC)=0.84×ΔNO$_x$ (ppbv)+22.16, ΔNMVOCs ($\mu$g m$^{-3}$)=2.04×ΔNO$_x$ ($\mu$g m$^{-3}$)+26.59.

[Figure]

**Figure 11.** Estimated annual vehicular emissions of (a) NO$_X$, (b) CO, and (c) VOCs in Tianjin during 2000-2016. The vehicular emission ratios (m/m) of VOCs to NO$_x$ in (d) Tianjin and (e) China. (PC: passenger cars; LDV: light-duty vehicles; HDT: heavy-duty trucks; BUS: buses; MC: motocycles. NAP: Standards are assumed at the same level of 2013 during the future years (e.g., China IV for light-duty gasoline vehicles and China III for heavy-duty diesel vehicls); PC1: Increasingly stringent emission standards and improved fuel quality are implemented according to a probably timetable; PC2: For internal combustion engine vehicles, the implementation of future emission standards is consistent with that under the PC1 scenario. The vehicle emissions of (a) NO$_x$, (b) CO, and (c) VOCs in Tianjin during 2000-2016 were estimated by the COPERT IV model. The estimated total vehicular emission ratios of VOCs to NO$_x$ in China for NAP, PC1, and PC2 scenarios from 2010 to 2030 were based on Wu et al. (2017)'s study.)

**Table 1.** Average emission factors of Wujinglu tunnel in this study and their comparisons with other tunnel studies.

| City | Tunnel | Gradient | Test year | Vehicle type | Emission factors (mg veh$^{-1}$ km$^{-1}$) | | | |
|------|--------|----------|-----------|--------------|------|------|--------|------|
| | | | | | NO | NO$_2$ | NO$_x$ | CO |
| Hong Kong[a] | ShingMun tunnel | 1% | 2004 | 50% DVs | N.A. | N.A. | 878±308 | 1845±434 |
| Taiwan[b] | Hsuehshan tunnel | -1.3% | 2006 | LDV | N.A. | N.A. | 145±67 | 910±470 |
| | | 1.3% | 2006 | LDV | N.A. | N.A. | 331±166 | 1470±630 |
| Shanghai[c] | East Yan'an Road tunnel | -3% | 2012 | 94.5% GVs | N.A. | N.A. | N.A. | 1266±889 |
| | | 3% | 2012 | 94.5% GVs | N.A. | N.A. | N.A. | 3353±2155 |
| Changsha[d] | Yingpan Road tunnel | -6% | 2013 | 96.3%-98.4% GVs | N.A. | N.A. | 121±22 | 754±561 |
| | | 6% | 2013 | 96.3%-98.4% GVs | N.A. | N.A. | 818±755 | 6050±5940 |
| Guangzhou[e] | Zhujiang tunnel | N.A. | 2013 | 88.2% GVs | N.A | N.A | 560±50 | N.A. |
| | | N.A. | 2014 | 61%LDV; 27% LPG; | N.A. | N.A. | 1286±204 | 3096±680 |
| | | | | 12% HDV | | | | |
| Shanghai[f] | East Yan'an Road tunnel | N.A. | 2016 | 94.1% GVs | N.A | N.A | 400±250 | 1840±900 |
| Tianjin[g] | Wujinglu tunnel | -4% | 2017 | 94.2% GVs | 62±72 | 17±11 | 79±78 | 270±342 |
| | | 4% | 2017 | 94.2% GVs | 159±73 | 24±20 | 181±88 | 578±382 |
| | | Overall | 2017 | 94.2% GVs | 98±70 | 16±9 | 117±78 | 345±250 |

[a] Hong Kong (Cheng et al., 2006); [b] Taiwan (Chang et al., 2008); [c] Shanghai (Deng et al., 2015); [d] Changsha (Deng et al., 2015); [e] Guangzhou (Liu et al., 2014; Zhang et al., 2015); [f] Shanghai (Huang et al., 2017); [g] This study.

N.A. – not available.

**Table 2.** Typical ratios (v/v) related to vehicle emissions compared to previous tunnel studies.

[revised manuscript text omitted]